# MoLF: Mixture-of-Latent-Flow for Pan-Cancer Spatial Gene Expression Prediction from Histology

**Susu Hu** [1 2 3 4]  **Stefanie Speidel** [1 2 3 4]

## Abstract

Inferring spatial transcriptomics (ST) from histology enables scalable histogenomic profiling, yet current methods are largely restricted to single-tissue models. This fragmentation fails to leverage biological principles shared across cancer types and hinders application to data-scarce scenarios. While pan-cancer training offers a solution, the resulting heterogeneity challenges monolithic architectures. To bridge this gap, we introduce **MoLF** (*Mixture-of-Latent-Flow*), a generative model for pan-cancer histogenomic prediction. MoLF leverages a conditional Flow Matching objective to map noise to the gene latent manifold, parameterized by a Mixture-of-Experts (MoE) velocity field. By dynamically routing inputs to specialized sub-networks, this architecture effectively decouples the optimization of diverse tissue patterns. Our experiments demonstrate that MoLF establishes a new state-of-the-art, consistently outperforming both specialized and foundation model baselines on pan-cancer benchmarks. Furthermore, MoLF exhibits zero-shot generalization to cross-species data, suggesting it captures fundamental, conserved histo-molecular mechanisms. Code is available at https://susuhu.github.io/MoLF/.

## 1. Introduction

Spatial transcriptomics (ST) enables high-resolution molecular profiling but remains prohibitively expensive and low-throughput (Khan et al., 2024). In contrast, H&E histology is ubiquitous. This disparity has motivated computational methods to infer ST directly from histology, enabling scalable histogenomic analysis.

Ideally, such inference should operate within a unified pan-cancer framework rather than relying on inefficient, tissue-specific models. A generalized approach can leverage fundamental biological processes conserved across malignancies. However, the extreme morphological heterogeneity of diverse tissues introduces conflicting learning signals, causing monolithic architectures to struggle with interference.

Methodologically, early approaches relied on deterministic regression (Xie et al., 2023; Zeng et al., 2022; Long et al., 2023; Chung et al., 2024). However, gene expression is inherently stochastic and multimodal: similar morphologies can correspond to diverse molecular states. This necessitates generative modeling to capture the full conditional distribution of gene expression.

Current generative strategies, however, face distinct limitations when scaled to pan-cancer analysis. Diffusion-based models, such as STEM (Zhu et al., 2025), rely on iterative denoising steps; their prohibitive computational cost and reliance on limited leave-one-out evaluation protocols hinder their deployment for robust, large-scale inference. Flow-matching approaches like STFlow (Huang et al., 2025a) alleviate the sampling bottleneck but rely on monolithic architectures. We hypothesize that without specialized mechanisms to resolve phenotypic heterogeneity, these models struggle to capture the conflicting morphological distributions inherent to pan-cancer data within a single shared parameter space. Meanwhile, foundation model STPath(Huang et al., 2025b) achieves scale via masked prediction. However, this approach requires intensive pre-training resources and optimizes a masked reconstruction objective rather than an explicitly generative one.

To bridge these gaps, we propose **MoLF**, a conditional flow matching framework with a Mixture-of-Experts (MoE) velocity parameterization. MoLF enables robust pan-cancer prediction by (1) structuring the latent gene space via a Variational Autoencoder (VAE), (2) modeling the conditional distribution via flow matching, and (3) dynamically routing inputs to specialized experts. This design explicitly

---

[1]Translational Surgical Oncology, National Center for Tumor Diseases (NCT/UCC) Dresden, Germany [2]Faculty of Medicine and University Hospital Carl Gustav Carus, Dresden University of Technology [3]Helmholtz-Zentrum Dresden-Rossendorf (HZDR), Dresden, Germany [4]German Cancer Research Center (DKFZ), Heidelberg, Germany. Correspondence to: Susu Hu <susu.hu@nct-dresden.de>.

*Proceedings of the 43rd International Conference on Machine Learning*, Seoul, South Korea. PMLR 306, 2026. Copyright 2026 by the author(s).

mitigates the parameter interference inherent in pan-cancer learning, allowing the model to capture both shared biological motifs and tissue-specific nuances.

## 2. Related Work

**Generative Modeling for Histogenomics.** Early computational pathology methods treated ST prediction as a deterministic regression task (Xie et al., 2023; Zeng et al., 2022; Long et al., 2023; Chung et al., 2024). A fundamental limitation of these approaches is the assumption of a one-to-one mapping between morphology and gene expression. In reality, gene expression is inherently stochastic and multimodal: visually similar histological patterns frequently correspond to diverse molecular profiles. Deterministic models are incapable of representing this one-to-many conditional distribution. To address this, generative approaches have emerged. Diffusion-based models, such as STEM (Zhu et al., 2025), capture this stochasticity but rely on computationally expensive iterative sampling. Flow-matching frameworks like STFlow (Huang et al., 2025a) accelerate inference via optimal transport paths but have primarily been restricted to single-cancer settings. MoLF advances this lineage by integrating Flow Matching with a Mixture-of-Experts architecture, combining sampling efficiency with the capacity to resolve complex, multi-tissue heterogeneity.

**Pan-Cancer Generalization.** Most existing ST predictors are trained on tissue-specific cohorts, limiting their applicability to data-scarce scenarios. Pan-cancer modeling aims to learn conserved histogenomic principles shared across malignancies. Currently, the leading approach in this domain is STPath (Huang et al., 2025b), a large-scale foundation model. However, STPath relies on a BERT-style masked prediction objective. While effective for representation learning, this is fundamentally a masked reconstruction task (inferring missing data from context) rather than a generative process that models the distribution from a stochastic prior. Furthermore, it requires data-intensive pre-training. In contrast, MoLF achieves robust pan-cancer generalization without massive pre-training by leveraging architectural specialization (MoE) to explicitly decouple and manage phenotypic diversity.

## 3. Methodology

We introduce **MoLF**, a two-stage generative framework designed to model the conditional distribution of spatially structured gene expression given histopathology images. The method first establishes a compressed, biologically meaningful latent manifold via a Variational Autoencoder (VAE) (Kingma et al., 2019), and subsequently learns a conditional probability flow to this manifold using a Mixture-of-Experts (MoE) velocity parameterization (Figure 1).

### 3.1. Stage I: Gene Manifold Learning via Transformer VAE

Let $x \in \mathbb{R}^{D_{\text{gene}}}$ denote a $log1$ normalized gene expression vector. We postulate a lower-dimensional latent variable $z \in \mathbb{R}^{D_{\text{latent}}}$ that captures coordinated biological programs. To approximate the posterior, we employ a Transformer-based VAE.

The probabilistic model is defined by the encoder $q_\phi(z|x)$ and decoder $p_\psi(x|z)$:

$$q_\phi(z|x) = \mathcal{N}(z; \mu_\phi(x), \text{diag}(\sigma_\phi^2(x))), \quad (1)$$

$$p_\psi(x|z) = \mathcal{N}(x; f_\psi(z), \sigma^2 I), \quad (2)$$

where $\mu_\phi, \sigma_\phi$ are parameterized by a Transformer encoder and $f_\psi$ by a Multi-Layer Perceptron (MLP). The parameters are optimized via the Evidence Lower Bound (ELBO):

$$\mathcal{L}_{\text{VAE}} = \mathbb{E}_{q_\phi}[\log p_\psi(x|z)] - \beta D_{\text{KL}}(q_\phi(z|x)\|\mathcal{N}(0, I)). \quad (3)$$

Upon convergence, $(\phi, \psi)$ are frozen. The latent space $\mathcal{Z}$ defines the target manifold for the subsequent generative stage.

### 3.2. Stage II: Conditional Flow Matching with MoE

We condition the generative process on a composite context vector $c = \{c_{\text{img}}, c_{\text{type}}\}$, where $c_{\text{img}}$ represents the histological image features and $c_{\text{type}}$ is the one-hot encoded cancer type. Given this context, our objective is to sample from the conditional distribution $p(z|c)$. We adopt Conditional Flow Matching (CFM) (Lipman et al., 2022), which learns a time-dependent vector field $v_\theta$ to transport a simple prior density $p_0$ to the data density $p_1 \approx p_{\text{data}}$.

#### 3.2.1. OPTIMAL TRANSPORT CONDITIONAL FLOW

We define a probability flow governed by the Ordinary Differential Equation (ODE):

$$\frac{dz_t}{dt} = v_\theta(z_t, t, c), \quad z_0 \sim \mathcal{N}(0, I), \quad (4)$$

where $t \in [0, 1]$. The vector field $v_\theta$ is parameterized by a neural network that integrates multimodal conditioning: histological image features $c_{\text{img}}$ are concatenated with the noisy state $z_t$, while the cancer type $c_{\text{type}}$ is injected via cross-attention layers.

To encourage straight trajectories, we utilize the Optimal Transport conditional vector field objective. For a training pair $(z_0, z_1)$, where $z_0$ is the source noise and $z_1$ is the target data, the interpolation path is defined as the geodesic:

$$\psi_t(z_0, z_1) = (1 - t)z_0 + tz_1. \quad (5)$$

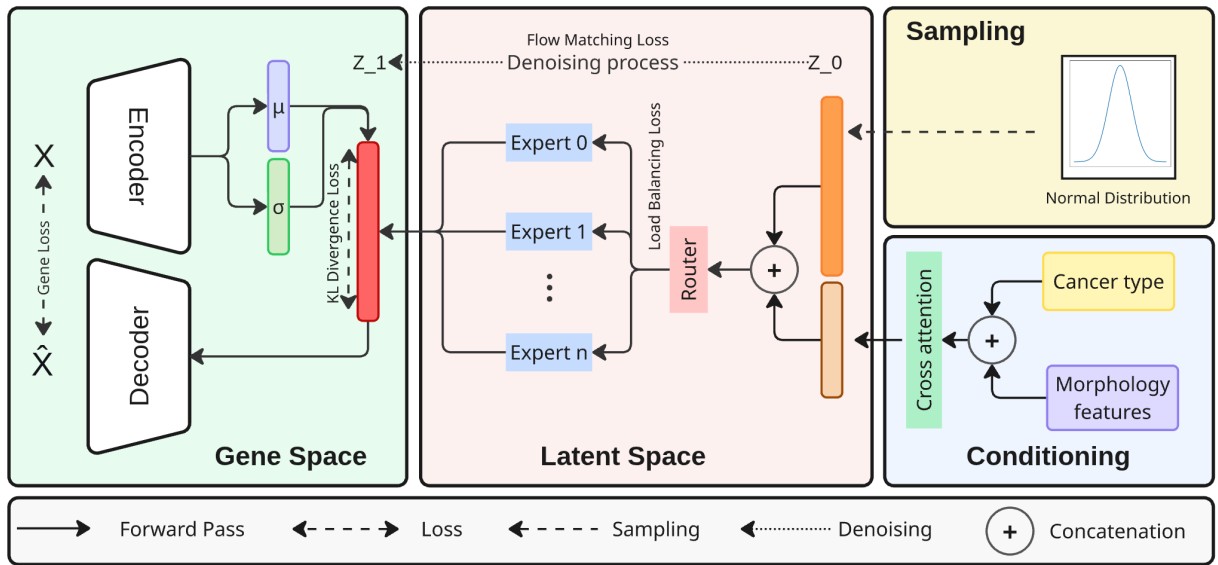

*Figure 1.* **Overview of MoLF Architecture. Stage I:** A Transformer-based VAE compresses high-dimensional gene expression into a structured latent manifold. **Stage II:** A Conditional Flow Matching model with Mixture-of-Experts (MoE) velocity estimation learns to transport noise to this latent manifold, conditioned on H&E morphological features.

The target vector field corresponds to the time derivative of this path, $u_t(z_0, z_1) = z_1 - z_0$. The primary objective minimizes the regression error between the network prediction and this target:

$$\mathcal{L}_{\text{CFM}}(\theta) = \mathbb{E}_{t,z_0,z_1} \left[ \|v_\theta(\psi_t, t, c) - u_t\|^2 \right]. \quad (6)$$

### 3.2.2. MIXTURE-OF-EXPERTS VELOCITY PARAMETERIZATION

Modeling pan-cancer heterogeneity requires a vector field $v_\theta$ capable of capturing diverse and potentially conflicting dynamic regimes. However, a single network with shared parameters struggles to simultaneously encode the distinct, and often conflicting, vector fields required for different tissue morphologies. To resolve this, we parameterize $v_\theta$ using a sparse Mixture-of-Experts (MoE) architecture.

**Dynamic Velocity Composition.** The velocity field is defined as a weighted sum of $N$ expert networks $\{E_i\}_{i=1}^N$:

$$v_\theta(z_t, t, c) = \sum_{i=1}^{N} G(z_t, t, c)_i \cdot E_i(z_t, t, c), \quad (7)$$

where $G(\cdot)$ is a gating network. We employ Top-$k$ gating, where only a subset $k \ll N$ experts are active for any given input, enabling the model to decompose the global transport map into local, expert-specific sub-problems. This effectively decouples the optimization landscapes for discordant tissue types.

### 3.2.3. REGULARIZATION AND TOTAL OBJECTIVE

**Biological Consistency Regularization.** To explicitly anchor the generative process to the gene expression manifold, we enforce a reconstruction consistency loss. We first estimate the terminal latent state $\hat{z}_1$ by projecting the current flow trajectory to $t = 1$:

$$\hat{z}_1 = z_t + (1 - t)v\theta(z_t, t, c). \quad (8)$$

We then decode this estimate using the frozen Stage I decoder $f_\psi$ and penalize the Mean Squared Error (MSE) against the ground truth gene expression $x$:

$$\mathcal{L}_{\text{gene}} = \|x - f\psi(\hat{z}_1)\|_2^2. \quad (9)$$

This ensures that the vector field directs samples toward regions of the latent space that correspond to high-fidelity, biologically valid transcriptomic profiles.

**Load Balancing.** To prevent expert collapse, we apply the auxiliary load-balancing loss $\mathcal{L}_{\text{aux}}$ proposed by Fedus et al. (Fedus et al., 2022). For a batch of $M$ samples, let $h_m \in \mathbb{R}^N$ denote the gating logits for the $m$-th sample. The routing probability distribution over the experts is given by the softmax function, $p_{m,i} = \frac{\exp(h_{m,i})}{\sum_{j=1}^{N} \exp(h_{m,j})}$.

We define $f_i$ as the fraction of the total routing probability allocated to expert $i$ across the batch:

$$f_i = \frac{1}{M} \sum_{m=1}^{M} p_{m,i} \quad (10)$$

Similarly, we define $P_i$ as the fraction of samples actually routed to expert $i$. Letting $\text{TopK}(h_m, k)$ denote the set of indices of the $k$ experts with the highest logits for sample $m$, the routing assignment is:

$$P_i = \frac{1}{M} \sum_{m=1}^{M} \mathbb{1}\{i \in \text{TopK}(h_m, k)\} \quad (11)$$

where $\mathbb{1}\{\cdot\}$ is the indicator function. The load-balancing loss is defined as the scaled dot product of these two distributions, which is minimized when samples are distributed uniformly across all $N$ experts:

$$\mathcal{L}_{\text{aux}} = N \sum_{i=1}^{N} f_i \cdot P_i \quad (12)$$

The total training objective is:

$$\mathcal{L}_{\text{total}} = \lambda_{\text{flow}}\mathcal{L}_{\text{CFM}} + \lambda_{\text{gene}}\mathcal{L}_{\text{gene}} + \lambda_{\text{aux}}\mathcal{L}_{\text{aux}}. \quad (13)$$

To enable Classifier-Free Guidance (CFG), we randomly drop the condition $c$ with probability $p_{\text{drop}} = 0.1$. We perform single-step Euler integration with guidance scale $w$: $\hat{v} = v_\theta(z_0, \emptyset) + w \cdot (v_\theta(z_0, c) - v_\theta(z_0, \emptyset))$. As increasing $w$ maximizes correlation but can degrade distributional realism (scale amplification), we employ a constrained "Filter-and-Rank" optimization protocol to select the optimal $w$ (details in Appendix C).

## 4. Experiments and Results

We evaluate our framework in three stages: (1) a controlled synthetic setting to assess conditional density estimation, (2) the primary benchmark of pan-cancer gene pathway prediction, and (3) ablation studies isolating architectural contributions.

### 4.1. Synthetic Assessment: Conditional Eight Gaussians

To isolate the inductive bias of the MoE architecture, we evaluate conditional density estimation on a synthetic 8-Gaussian task. We utilized $N = 8$ experts with a Top-1 gating strategy to match the distinct modes of the data distribution (details in Appendix A.3). We compare our MoE-Transformer against a Dense Transformer baseline of identical depth and dimension.

**Shared vs. Specialized Parameterization.** The Dense Baseline relies on a global set of parameters to map all eight conditioning variables to their spatially disjoint modes. This shared parameterization forces the model to find a compromise representation that accommodates competing transport directions, often resulting in averaged or "smeared" outputs. In contrast, MoLF explicitly decomposes the problem: the

router assigns modes to distinct experts, allowing the model to learn eight local transport plans without the burden of fitting a single global function.

Results in Figure 2 and Table 1 support this. The Dense Baseline struggles to cleanly separate the modes, exhibiting artifacts where the model interpolates between targets. MoLF achieves tighter concentration, confirming that decoupling the experts simplifies the learning of multi-modal distributions.

*Table 1.* **Quantitative Results of Synthetic Gaussian.** The MoE architecture yields lower 2-Wasserstein distance ($W_2$) compared to the Dense Baseline.

| MODEL | DIM-1↓ | DIM-2↓ | AVERAGE↓ |
|---|---|---|---|
| MOLF (OURS) | **0.315** | **0.268** | **0.292** |
| MOLF - W/O MOE | 0.358 | 0.272 | 0.315 |

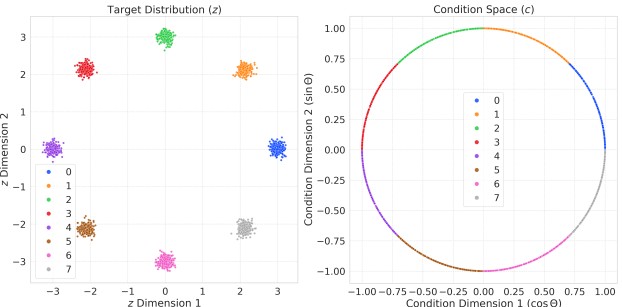

*(a)* **Ground Truth:** Target distribution (left) and condition distribution (right).

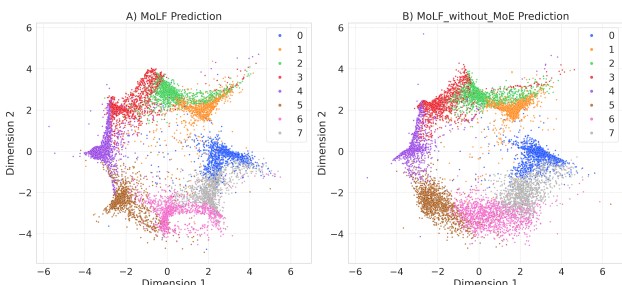

*(b)* **Prediction Comparison:** MoLF (left) generates tightly concentrated modes. The Dense Baseline (right) suffers from higher variance and mode-connection artifacts.

*Figure 2.* **Qualitative Results of Synthetic Gaussian.** By routing conditions to specialized experts, the MoE architecture avoids the averaging artifacts seen in the dense baseline.

### 4.2. Pan-Cancer Gene Expression Prediction

#### 4.2.1. EXPERIMENTAL SETUP AND BENCHMARKS

We conduct comprehensive experiments on the HEST-1k pan-cancer benchmark (Jaume et al., 2024), adhering to the official splits to ensure rigorous held-out evaluation.

This large-scale collection encapsulates significant inter-center variability, spanning diverse spatial transcriptomics platforms, staining protocols, and scanner vendors, thereby ensuring that our results reflect true cross-site generalization. For training we include all available slides covering 29 cancer types including "unknown". We encode histological morphology using visual features extracted from the UNI-v2 pathology foundation model (Chen et al., 2024), ensuring a consistent feature space for MoLF and the retrained baselines. We utilized $N = 6$ experts with a Top-2 gating strategy ($k = 2$). We adopted this configuration to ensure training stability and maximize expert utilization; Top-2 routing mitigates the risk of routing collapse often associated with hard Top-1 strategies while providing sufficient capacity to model pan-cancer heterogeneity.

**Curated Gene Panel.** A significant challenge in histogenomic modeling is defining a predictive target that balances biological significance with rigorous evaluation. While recent foundation model STPath (Huang et al., 2025b) attempted to predict expansive panels (up to 38k genes); however, without assessing performance on the long tail of low-variance genes, this coverage provides no guaranteed predictive value. To establish a robust benchmark, we construct a curated panel comprising the union of the 50 MSigDB Hallmark pathways (Broad Institute, 2025) and the top-50 cancer-specific Highly Variable Genes (HVG). This set is designed to test two distinct capabilities: modeling stable, conserved biological programs (Hallmark) and capturing strong, cancer-specific variance signals (HVG). Detailed statistics are provided in Appendix A.1.

**Baselines.** We benchmark MoLF against a comprehensive suite of architectural classes representing the current state-of-the-art, which have recently surpassed previous regression methods (Pang et al., 2021; He et al., 2020; Xie et al., 2023; Zeng et al., 2022; Long et al., 2023; Chung et al., 2024) in their respective benchmarks. We include the diffusion-based **STEM** (Zhu et al., 2025) and flow-based **STFlow** (Huang et al., 2025a) as generative baselines, alongside the large-scale BERT-style foundation model **STPath** (Huang et al., 2025b) to represent the foundation model paradigm. Additionally, we employ a standard **MLP** to serve as a deterministic anchor. We applied a unified experimental protocol to all retrained baselines (MLP, STFlow, STEM). However, for STEM, we were forced to restrict the experiments with only the HVG gene panel, as scaling the iterative diffusion process to the full pathway space was computationally prohibitive. STPath was evaluated using its official pre-trained checkpoint.

#### 4.2.2. QUANTITATIVE PERFORMANCE

**Performance on Highly Variable Genes.** We first evaluate performance on the Top-50 HVG (Table 2), reporting the

*Table 2.* Top50 HVG PCC ↑ across 10 cancer types. Best in **bold**.

| Cancer Type | MLP | STPath | STFlow | STEM | MoLF (ours) |
|---|---|---|---|---|---|
| CCRCC | $0.194_{0.012}$ | $0.117_{0.001}$ | $0.128_{0.037}$ | $0.096_{0.028}$ | $\mathbf{0.231}_{0.065}$ |
| COAD | $0.343_{0.114}$ | $0.393_{0.185}$ | $0.310_{0.002}$ | $0.235_{0.076}$ | $\mathbf{0.406}_{0.167}$ |
| HCC | $0.064_{0.045}$ | $0.094_{0.052}$ | $0.081_{0.047}$ | $0.052_{0.015}$ | $\mathbf{0.101}_{0.045}$ |
| IDC | $0.561_{0.125}$ | $\mathbf{0.629}_{0.126}$ | $0.518_{0.078}$ | $0.376_{0.095}$ | $0.628_{0.124}$ |
| LUNG | $0.507_{0.030}$ | $0.518_{0.028}$ | $0.465_{0.040}$ | $0.235_{0.001}$ | $\mathbf{0.525}_{0.041}$ |
| LYMPH_IDC | $0.217_{0.071}$ | $0.182_{0.075}$ | $0.189_{0.045}$ | $0.079_{0.020}$ | $\mathbf{0.245}_{0.060}$ |
| PAAD | $0.407_{0.122}$ | $\mathbf{0.493}_{0.100}$ | $0.416_{0.122}$ | $0.283_{0.151}$ | $0.460_{0.084}$ |
| PRAD | $0.335_{0.065}$ | $0.257_{0.012}$ | $0.227_{0.005}$ | $0.197_{0.074}$ | $\mathbf{0.348}_{0.003}$ |
| READ | $0.264_{0.040}$ | $\mathbf{0.280}_{0.030}$ | $0.226_{0.056}$ | $0.168_{0.013}$ | $0.275_{0.045}$ |
| SKCM | $0.515_{0.104}$ | $0.588_{0.113}$ | $0.557_{0.087}$ | $0.237_{0.139}$ | $\mathbf{0.606}_{0.127}$ |
| **Average** | $0.341_{0.160}$ | $0.361_{0.072}$ | $0.312_{0.168}$ | $0.196_{0.004}$ | $\mathbf{0.382}_{0.173}$ |

mean and standard deviation across two independent training splits (details in Appendix A). As these genes possess high variance, they typically offer strong supervision signals. In this regime, MoLF achieves state-of-the-art performance, outperforming all baselines. This suggests that MoLF's conditional flow-matching framework effectively resolves the ill-posed inverse problem intrinsic to ST prediction, where similar histological morphologies may correspond to diverse molecular states.

Notably, the diffusion-based STEM yields suboptimal performance in this setting, notwithstanding the theoretical advantage of operating on a restricted, high-variance gene panel. While Zhu et al. (2025) reported strong results using a leave-one-out evaluation, the significant performance drop observed here raises critical questions regarding the external validity of that protocol. Our rigorous fixed hold-out split reveals STEM's struggle to generalize across heterogeneous pan-cancer domains.

**Performance on Hallmark Pathways.** To probe model capabilities beyond dominant variance signals, we evaluate performance on the Hallmark pathway genes, stratified into three tiers (Low, Mid, High) based on average gene variance (Table 3).

MoLF achieves the highest average Pearson Correlation Coefficient (PCC) across all variance tiers. Interestingly, while STPath remains a competitive baseline on HVG, demonstrating the efficacy of masked-prediction for dominant domain signals, it underperforms the simple MLP on stratified pathway genes. This indicates that BERT-style modeling may capture high-level statistics but misses the nuanced, lower-variance signals required for comprehensive pathway analysis. Conversely, MoLF's mixture-of-latent-flows design explicitly models pan-cancer heterogeneity in a better aligned gene latent space, allowing it to significantly outperform the single-flow STFlow baseline.

### 4.3. Geometric Analysis of Generative Transport

We qualitatively validate the geometric properties of our flow matching framework by visualizing the transformation from the prior distribution to the data manifold. We project

*Table 3.* Model performance comparison measured in PCC of variance-stratified hallmark genes. Best in **bold**.

| MODEL | LOW-VAR ↑ | MID-VAR ↑ | HIGH-VAR ↑ | OVERALL AVG. ↑ |
|---|---|---|---|---|
| MLP | $0.225_{0.028}$ | $0.144_{0.018}$ | $0.145_{0.143}$ | $0.171_{0.045}$ |
| STFLOW | $0.158_{0.011}$ | $0.100_{0.007}$ | $0.107_{0.003}$ | $0.122_{0.029}$ |
| STPATH | $0.177_{0.016}$ | $0.113_{0.011}$ | $0.137_{0.013}$ | $0.143_{0.031}$ |
| MoLF (OURS) | $\mathbf{0.242}_{0.003}$ | $\mathbf{0.159}_{0.001}$ | $\mathbf{0.167}_{0.006}$ | $\mathbf{0.185}_{0.002}$ |

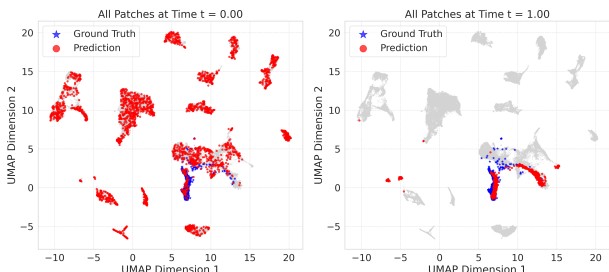

*Figure 3.* **Macro-scale generative transport. Left**: Initial state ($t = 0$), where the prediction is randomly initialed Gaussian noise, projected in UMAP space (red). **Right**: After one ODE step, this noise is transformed into a structured manifold (red) that aligns with the sample's ground truth distribution (blue), confirming global distributional matching.

the latent integration paths into a 2D UMAP space fitted on the ground truth test set latents.

**Macro-scale Manifold Alignment.** Figure 3 illustrates the global transport of a sample's distribution. The process initializes at $t = 0$ with isotropic Gaussian noise warped into learned latent gene manifold (left). Following a single ODE solver step guided by the histology-conditioned velocity field, the noise is transported to a structured configuration at $t = 1$ (right). The generated distribution aligns closely with the topological structure of the ground truth gene manifold, demonstrating that the model successfully approximates the target density in a single forward pass.

**Micro-scale Conditional Coupling.** To verify that the model learns specific input-output pairings rather than a generic mean mapping, we trace individual patch trajectories in Figure 4. For randomly selected patches, the flow maps the initial noise (green dot) to a terminal point (red cross) that is highly spatially aligned with the specific ground truth embedding (cyan star). This indicates the model learns a precise, distinct velocity field for each patch, effectively solving the optimal transport problem between the noise and data distributions.

### 4.4. Expert Specialization and Utilization

To understand how the MoE router handles pan-cancer heterogeneity, we analyze expert utilization at $t = 0$. At this initialization step, the gating mechanism operates on the

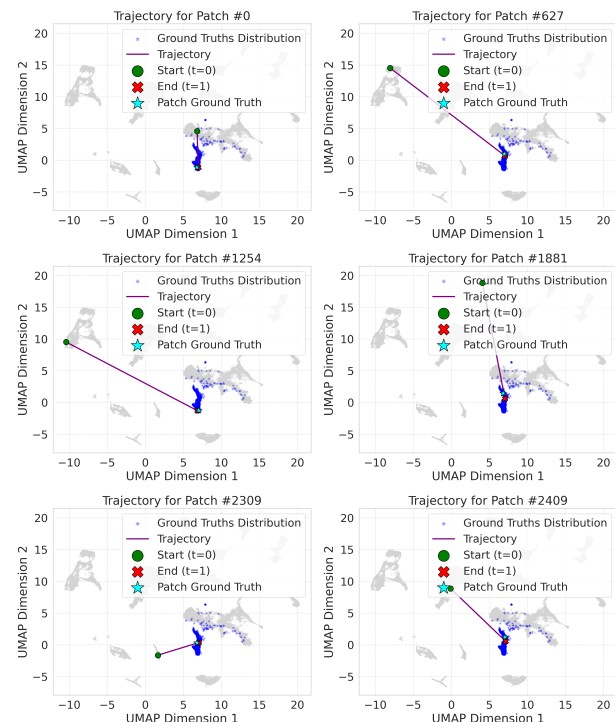

*Figure 4.* **Micro-scale trajectory analysis.** Trajectories for six random patches from a single sample. The model transports a random starting noise vector (green dot) to a predicted latent (red 'X') via a single straight-line step. The prediction lands in close proximity to the specific ground truth destination (cyan star), demonstrating accurate conditional coupling.

starting Gaussian noise conditioned on H&E morphological features, effectively determining the trajectory onset of the flow matching process.

**Distributed Representation Strategy.** We visualize the top-activating patches for each expert and their corresponding gene latent embeddings (Figure 5). Contrary to a "class-specialist", we observe no distinct partitioning of experts by cancer type. As shown in the UMAP visualization, expert activations are distributed across the latent space rather than clustered by disease label.

Table 4 details the routing distribution across cancer types. While a load-balancing loss ensures non-collapse during training, inference reveals a collaborative encoding strategy. Table 4 shows that activation patterns vary across cancer types, reflecting a collaborative, distributed encoding strategy. This distributed processing likely supports generalization by learning feature-level invariances rather than dataset-specific correlations. Experiment in Section 4.5 supports this assumption.

Further analysis of how expert specialization dynamically

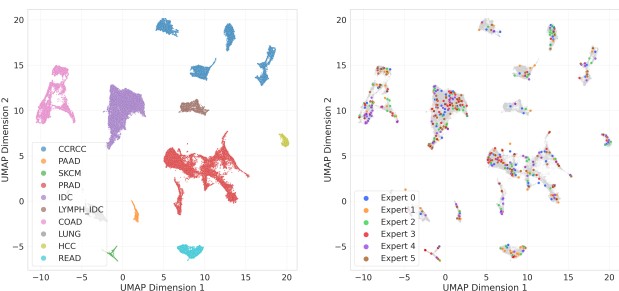

*Figure 5.* **Expert specialization analysis. Left**: Gene latent embeddings colored by cancer type. **Right**: The same gene latent embeddings overlaid with activated colored experts. The lack of distinct clustering of experts in the right panel indicates that experts are not segregated by cancer type, but rather contribute collaboratively across the dataset.

*Table 4.* Expert routing distribution per cancer type (%). Highest is in **bold**, second is with underlined. The distributed activation confirms a collaborative MoE strategy.

| CANCER | E1 | E2 | E3 | E4 | E5 | E6 |
|---|---|---|---|---|---|---|
| CCRCC | 19.26 | 17.08 | 19.55 | 7.97 | **19.91** | 16.22 |
| COAD | **19.72** | 18.49 | 17.65 | 10.86 | 16.87 | 16.42 |
| HCC | 18.14 | 18.65 | 17.50 | 5.93 | **23.18** | 16.61 |
| IDC | 18.42 | 10.90 | **19.78** | 15.00 | 17.75 | 18.15 |
| LUNG | 17.42 | 14.96 | 17.35 | 13.69 | **20.25** | 16.34 |
| LYMPH_IDC | 17.58 | 10.28 | 19.14 | 12.05 | **23.39** | 17.56 |
| PAAD | 17.60 | **20.67** | 20.36 | 7.50 | 18.40 | 15.47 |
| PRAD | 18.33 | 15.53 | **19.58** | 14.05 | 16.07 | 16.43 |
| READ | 18.03 | **27.60** | 20.17 | 9.43 | 11.39 | 13.38 |
| SKCM | 17.39 | **18.45** | 17.66 | 13.53 | 17.98 | 14.99 |

adapts to target gene panel, shifting from functional alignment to structural resolution, is detailed in the ablation study (Appendix D.1).

### 4.5. Zero-Shot Cross-Species Generalization

To assess robustness to domain shift, we evaluate the models on a zero-shot cross-species task. All models were trained on human data and evaluated on the HEST1k mouse melanoma dataset with cancer type as "unknown". This evaluation is biologically motivated by foundational research establishing that genetically engineered mouse models can effectively recapitulate the critical genetic drivers and histo-morphological patterns of human melanoma (Dankort et al., 2009; Patton et al., 2021).

Table 5 presents the quantitative results. MoLF demonstrates superior out-of-distribution (OOD) robustness, outperforming both regression and generative baselines. Notably, the foundation model STPath shows limited transfer capability, suggesting that while large-scale masked modeling captures in-distribution statistics effectively, it may overfit to species-specific artifacts rather than learning invariant biological signals.

We further investigate the impact of training data diversity

*Table 5.* Zero-shot cross-species inference on mouse melanoma. Performance measured in PCC ↑. Best in **bold**.

| MODEL | PCC ↑ |
|---|---|
| MLP | $0.145_{0.009}$ |
| STPATH | $0.090_{0.000}$ |
| STFLOW | $0.126_{0.023}$ |
| MOLF - SKIN CANCER ONLY | $0.127_{0.047}$ |
| MOLF (OURS) | $\mathbf{0.202}_{0.001}$ |

by benchmarking a variant of our model trained exclusively on human skin cancer ("MoLF - Skin Cancer Only"). Consistent with the central hypothesis of this work, the pan-cancer MoLF significantly outperforms this tissue-aligned baseline. This validates our motivation that restricting training to specific cancer types limits the model's ability to learn robust, transferable features. Instead, the diversity of the pan-cancer dataset enables the model to learn broader, species-invariant morphological motifs that generalize more effectively than models trained on tissue-specific subsets.

These findings support the conclusion that the MoE architecture, combined with diverse pan-cancer training, facilitates the decomposition of visual signals into fundamental biological primitives. By learning reusable expert representations, MoLF avoids overfitting to human-specific histology and maintains predictive stability on unseen cross-species data.

### 4.6. Ablation Study

#### 4.6.1. THE ROLE OF THE MIXTURE-OF-EXPERTS

To isolate the contribution of our Mixture-of-Experts (MoE) architecture, we trained an ablation model where the MoE velocity predictor was replaced with a standard attention velocity predictor. This allows us to compare the effectiveness of two different strategies for handling the complexity of pan-cancer data. Monolithic models inherently entangle disparate histological concepts within a single weight space. Our MoE architecture explicitly decouples these signals, routing them to distinct experts to avoid capacity bottlenecks.

Our full model with MoE consistently and significantly outperforms the ablation model across all cancer types and gene variance levels as shown in Table 6 and Table 7, demonstrating the superiority of the MoE approach.

#### 4.6.2. HOW DOES SPATIAL AWARENESS MATTER?

To investigate the impact of spatial structure, we ablated the sinusoidal Positional Encoding (PE) from the initial self-attention layer on H&E image features, specifically prior to the cross-attention injection of cancer type embeddings (Table 8 and Table 9). Removing PE, effectively treating the slide as a "bag-of-patches", yields slightly better per-

*Table 6.* The effect of MoE architecture in MoLF measured in Top50 HVG PCC ↑ across 10 cancer types. Best in **bold**.

| CANCER TYPE | MoLF | MoLF - W/O MoE |
|---|---|---|
| CCRCC | $\mathbf{0.231}_{0.063}$ | $0.159_{0.068}$ |
| COAD | $\mathbf{0.415}_{0.177}$ | $0.204_{0.088}$ |
| HCC | $\mathbf{0.100}_{0.044}$ | $0.041_{0.006}$ |
| IDC | $\mathbf{0.624}_{0.115}$ | $0.378_{0.016}$ |
| LUNG | $\mathbf{0.496}_{0.083}$ | $0.364_{0.021}$ |
| LYMPH_IDC | $\mathbf{0.243}_{0.053}$ | $0.169_{0.049}$ |
| PAAD | $\mathbf{0.484}_{0.067}$ | $0.135_{0.061}$ |
| PRAD | $\mathbf{0.356}_{0.000}$ | $0.270_{0.017}$ |
| READ | $\mathbf{0.278}_{0.045}$ | $0.093_{0.121}$ |
| SKCM | $\mathbf{0.605}_{0.118}$ | $0.279_{0.033}$ |
| AVERAGE | $\mathbf{0.383}_{0.172}$ | $0.209_{0.112}$ |

*Table 7.* The effect of MoE architecture in MoLF measured in PCC of variance-stratified hallmark genes. Best in **bold**.

| MODEL | LOW-VAR ↑ | MID-VAR ↑ | HIGH-VAR ↑ | OVERALL AVG. ↑ |
|---|---|---|---|---|
| MoLF - W/O MoE | $0.184_{0.002}$ | $0.114_{0.005}$ | $0.098_{0.009}$ | $0.127_{0.004}$ |
| MoLF | $\mathbf{0.242}_{0.003}$ | $\mathbf{0.159}_{0.001}$ | $\mathbf{0.167}_{0.006}$ | $\mathbf{0.185}_{0.002}$ |

*Table 8.* The effect of PE in MoLF measured in Top50 HVG PCC ↑ across 10 cancer types. Best in **bold**.

| CANCER TYPE | MoLF | MoLF - W/O PE |
|---|---|---|
| CCRCC | $\mathbf{0.231}_{0.063}$ | $0.181_{0.008}$ |
| COAD | $\mathbf{0.415}_{0.177}$ | $0.395_{0.018}$ |
| HCC | $0.100_{0.044}$ | $\mathbf{0.107}_{0.025}$ |
| IDC | $\mathbf{0.624}_{0.115}$ | $0.621_{0.108}$ |
| LUNG | $0.496_{0.083}$ | $\mathbf{0.560}_{0.043}$ |
| LYMPH_IDC | $\mathbf{0.243}_{0.053}$ | $0.242_{0.077}$ |
| PAAD | $0.484_{0.067}$ | $\mathbf{0.511}_{0.068}$ |
| PRAD | $\mathbf{0.356}_{0.000}$ | $0.331_{0.042}$ |
| READ | $\mathbf{0.278}_{0.045}$ | $0.250_{0.009}$ |
| SKCM | $0.605_{0.118}$ | $\mathbf{0.667}_{0.080}$ |
| AVERAGE | $0.383_{0.172}$ | $\mathbf{0.386}_{0.195}$ |

*Table 9.* The effect of PE in MoLF measured in PCC of variance-stratified hallmark genes. Best in **bold**.

| MODEL | LOW-VAR ↑ | MID-VAR ↑ | HIGH-VAR ↑ | OVERALL AVG. ↑ |
|---|---|---|---|---|
| MoLF - W/O PE | $0.222_{0.029}$ | $0.144_{0.022}$ | $0.159_{0.024}$ | $0.173_{0.018}$ |
| MoLF | $\mathbf{0.242}_{0.003}$ | $\mathbf{0.159}_{0.001}$ | $\mathbf{0.167}_{0.006}$ | $\mathbf{0.185}_{0.002}$ |

formance on Highly Variable Genes (HVG). This suggests that HVG expression is primarily driven by local patch morphology rather than spatial location. However, this lack of spatial structure consistently degrades Hallmark Pathway prediction, confirming that these biological processes rely on broader regional tissue architecture.

This distinction elucidates the performance hierarchy on structure-dependent pathways, where MoLF > MLP > STPath > STFlow. STFlow and STPath utilize Frame Averaging (FA) to enforce $E(2)$-invariance. This a rigid geometric bias ties representations to relative orientation while discarding the absolute coordinates necessary for regional pathway coherence. Furthermore, relying on local neighbor propagation to capture long-range context suffers from signal attenuation, as irrelevant intermediate tissue acts as an information bottleneck. Consequently, the geometry-aware STPath underperforms even the geometry-agnostic MLP baseline on pathway genes. In contrast, by leveraging a Mixture-of-Experts velocity parameterization, MoLF breaks the "one-size-fits-all" geometric constraint. The architecture provides the sufficient degrees of freedom to decouple the optimization of texture-driven and structure-driven objectives, allowing the model to reconcile these conflicting requirements where monolithic baselines fail.

## 5. Conclusion

In this work, we introduced MoLF, a conditional flow matching framework designed for pan-cancer histogenomic inference. By integrating a conditional flow matching with a Mixture-of-Experts (MoE) velocity parameterization at gene latent space, MoLF successfully resolves the conflicting optimization signals inherent in pan-cancer modeling.

Our extensive evaluations demonstrate that MoLF establishes a new state-of-the-art, outperforming both regression and generative baselines on high-variable genes (HVG) prediction and stratified Hallmark pathway coherence. Beyond in-distribution accuracy, the framework exhibits robust zero-shot generalization to cross-species data. This emergent capability suggests that by dynamically routing inputs to specialized experts, MoLF learns to decompose histology into fundamental, reusable morphological primitives that are conserved across species, rather than overfitting to dataset-specific artifacts.

**Limitations and Future Work.** While our analysis confirms that expert collaboration drives performance, the specific morphological criteria used by the gating network remain implicit. Currently, we observe *that* experts specialize, but we lack a verbal description of *what* concept each expert represents. Future work will focus on bridging this interpretability gap, potentially by leveraging pathology Vision-Language Models (VLMs) to automatically caption expert-activating patches or by incorporating pathologist-in-the-loop evaluation to ground these learned primitives in clinical semantics.

## Acknowledgement

This work is partly supported by the Federal Ministry of Research, Technology and Space in DAAD project 57616814 (SECAI, School of Embedded Composite AI, https://secai.org/) as part of the program Konrad Zuse Schools of Excellence in Artificial Intelligence.

## Impact Statement

This paper presents work whose goal is to advance the field of machine learning by enabling scalable, pan-cancer spatial transcriptomics prediction from histology images. While our generative model promises to democratize molecular profiling, it provides probabilistic predictions and must not be used for direct clinical decision-making without rigorous prospective validation. Furthermore, addressing potential training data biases and the interpretability gap of our Mixture-of-Experts architecture remain critical requirements for responsible real-world deployment.

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

# A. Appendix: Dataset

## A.1. Spatial Transcriptomics Dataset HEST1k

The detailed number of samples of two splits from HESK1k is show in table 10.

*Table 10.* Number of samples in splits.

| Splits | #Train | #Validation | #Test |
|--------|--------|-------------|-------|
| Split 0 | 385 | 96 | 23 |
| Split 1 | 380 | 96 | 28 |

## A.2. Gene Panel Details

The union of HVG and Hallmark pathway genes results in a gene panel of size 1386. To ground our analysis, we characterized the statistical properties of our gene sets. As shown in Table 11, the HVG set exhibits a higher mean and median variance than the Hallmark gene set, confirming our panel contains a diverse mix of signal types. Furthermore, we stratify the Hallmark genes into three tiers (Low-, Mid-, High-Variance), detailed in Table 12. This stratification enables a fine-grained assessment of a model's ability to capture the subtle but functionally critical pathways that large, unfiltered gene panels often obscure, providing a deeper insight into its true capabilities.

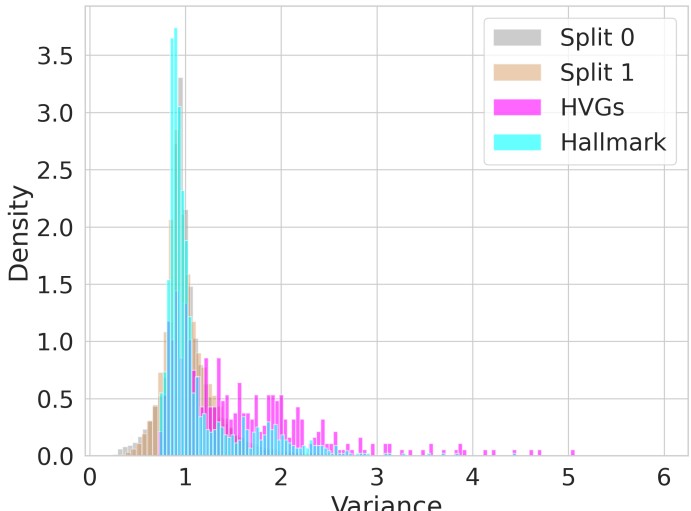

*Figure 6.* Gene variance distribution of test sets.

## A.3. Conditional Eight Gaussian Dataset

The data visualized in Figure 2a is generated as follows. First, an angle $\theta \sim \mathcal{U}(0, 2\pi)$ is sampled, and the condition is set to $c = [\cos(\theta), \sin(\theta)]^T$. This angle determines a discrete mode index $k = \lfloor 4\theta/\pi \rfloor \in \{0, \ldots, 7\}$. The target sample $z$ is then drawn from the corresponding Gaussian mode, $z \sim \mathcal{N}(\mu_k, \sigma^2 I)$. The eight means, $\mu_k$, are arranged in a ring of radius $R$, with $\mu_k = [R\cos(k\pi/4), R\sin(k\pi/4)]^T$. For our experiments, we use $R = 3.0$ and $\sigma^2 = 0.01$. The task for the model is to learn this mapping from a continuous condition to a multi-modal distribution.

For the conditional flow matching model with MoE, we use top-1 activation of 8 experts given this know eight-gaussian distribution. Each expert is implemented as a single-layer, single-head transformer with a latent dimension of 32. For the dense transformer, we use a single layer with 8 heads and a latent dimension of $8 \times 32 = 256$.

*Table 11.* Variance statistics for all gene groups.

| GENE SET | MEAN VARIANCE | MEDIAN VARIANCE | STD. VARIANCE |
|---|---|---|---|
| HVG | 1.540 | 1.345 | 0.729 |
| HALLMARK | 1.143 | 0.955 | 0.458 |
| ALL GENES | 1.204 | 0.973 | 0.547 |

*Table 12.* Hallmark genes variance thresholds used to define low-, mid-, and high-variance gene groups.

| VARIANCE LEVEL | MIN VARIANCE | MAX VARIANCE |
|---|---|---|
| LOW | 0.3998 | 0.9178 |
| MID | 0.9178 | 1.0209 |
| HIGH | 1.0211 | 4.1918 |

## B. Appendix: Implementation Details

We implement MoLF in PyTorch and utilize the UNI-v2 foundation model for visual feature extraction. The Stage I Gene VAE is constructed as a single-layer Transformer with 4 attention heads and a hidden dimension of 512, which compresses the gene expression input into a latent dimension of 128. We optimize this stage for 1000 epochs using the AdamW optimizer with a learning rate of $5 \times 10^{-5}$.

The Stage II conditional flow matching model operates with a hidden dimension of 256. The velocity field is parameterized by a Mixture-of-Experts architecture consisting of 6 experts with Top-2 routing. Each expert is defined as a single-layer Transformer with 4 heads and a dimension of 256. We train this stage for 500 epochs using a differential learning rate strategy, setting the backbone learning rate to $5 \times 10^{-5}$ and the gating network learning rate to $1 \times 10^{-5}$ to ensure routing stability. The total objective balances the flow matching loss and the auxiliary load-balancing loss with equal weighting ($\lambda_{\text{flow}} = 1.0, \lambda_{\text{gene}} = 1.0, \lambda_{\text{aux}} = 1.0$). We enable Classifier-Free Guidance by applying a condition dropout rate of 0.1 during training. Both Stage I and II are trained with early stopping patience of 50 epochs. Training is done on a single A100 GPU.

For all baseline methods, including STEM, STFlow, and STPath, we utilized the official implementations provided in their respective GitHub repositories and strictly adhered to their recommended hyperparameter configurations. Code is available at https://github.com/susuhu/MoLF.

## C. Appendix: Inference Hyperparameter Selection

We employ a constrained optimization protocol to select the Classifier-Free Guidance (CFG) scale $w$. A primary challenge in evaluating flow-matched transcriptomic profiles is that correlation-based metrics (e.g., Pearson correlation) are scale-invariant. As guidance strength increases ($w \gg 1$), the model tends to amplify conditional signal magnitudes. While this maximizes rank-order correlation, it often degrades predictive utility by generating distributions that diverge significantly from biological reality (high magnitude error).

To avoid selecting these "hallucinated" high-correlation models, we decouple pattern alignment from distributional realism using a *Filter-and-Rank* strategy. Let $\mathbf{y}_i$ be the ground truth expression vector for spot $i$ and $\hat{\mathbf{y}}_{i,w}$ be the generated profile at scale $w$. We define distributional divergence via the 1-Wasserstein metric ($\mathcal{D}_{W_1}$) and pattern error via Cosine distance ($\mathcal{D}_{\cos}$). For reproducibility, $\mathcal{D}_{W_1}$ is computed as the mean 1-Wasserstein distance between the distributions of expression values within each spot, averaged across all $N$ spots in the validation set: $\mathcal{D}_{W_1} = \frac{1}{N} \sum_i W_1(\mathbf{y}_i, \hat{\mathbf{y}}_{i,w})$.

**Step 1: Filter.** We first establish a baseline for biological realism by identifying the minimal distributional error observed across the sweep, $E^* = \min_w \mathcal{D}_{W_1}(\mathbf{y}, \hat{\mathbf{y}}_w)$. We define a feasible set of "valid" scales, $\mathcal{S}_{\text{valid}}$, containing models that remain within a distributional tolerance $\tau = 0.05$ of the baseline:

$$\mathcal{S}_{\text{valid}} = \{w \in \mathcal{W} \mid \mathcal{D}_{W_1}(\mathbf{y}, \hat{\mathbf{y}}_w) \leq (1 + \tau)E^*\}. \tag{14}$$

**Step 2: Rank.** From this biologically realistic subset, we select the optimal scale $w^*$ that maximizes pattern fidelity:

$$w^* = \underset{w \in \mathcal{S}_{\text{valid}}}{\arg\min} \, \mathcal{D}_{\cos}(\mathbf{y}, \hat{\mathbf{y}}_w). \tag{15}$$

This ensures the selected model captures sharp biological signals without succumbing to the scale-amplification artifacts typical of high-guidance sampling.

In Table 13, we present the generation quality metrics across varying guidance scales. Based on our *Filter-and-Rank* selection protocol, we selected a guidance scale of $w = 3.0$ for Split 0 and $w = 2.0$ for Split 1, as these configurations maximize pattern fidelity while maintaining distributional plausibility.

*Table 13.* **Guidance Scale Sensitivity.** Comparison of distributional metrics across validation splits. Selection involves a trade-off: lower $w$ favors distributional realism (Wasserstein), while moderate $w$ favors pattern matching (Cosine). Best results per column in **bold**.

| | SPLIT 0 | | | SPLIT 1 | | |
|---|---|---|---|---|---|---|
| **CFG ($w$)** | **MSE↓** | **WASS.↓** | **COS.↓** | **MSE↓** | **WASS.↓** | **COS.↓** |
| 1.0 | 0.205 | 0.162 | 0.241 | **0.191** | 0.157 | 0.255 |
| 2.0 | 0.190 | 0.161 | 0.233 | 0.194 | **0.157** | **0.253** |
| 3.0 | **0.189** | 0.160 | **0.233** | 0.203 | 0.162 | 0.255 |
| 4.0 | 0.189 | 0.158 | 0.233 | 0.215 | 0.171 | 0.259 |
| 5.0 | 0.191 | 0.156 | 0.234 | 0.225 | 0.172 | 0.263 |
| 6.0 | 0.193 | **0.155** | 0.235 | 0.233 | 0.181 | 0.266 |
| 7.0 | 0.196 | 0.162 | 0.235 | 0.239 | 0.186 | 0.269 |
| 8.0 | 0.199 | 0.164 | 0.236 | 0.243 | 0.183 | 0.271 |
| 9.0 | 0.202 | 0.162 | 0.237 | 0.246 | 0.182 | 0.272 |
| 10.0 | 0.204 | 0.167 | 0.238 | 0.248 | 0.189 | 0.273 |

# D. Appendix: Extended Analysis of Expert Specialization

## D.1. The Regularizing Effect of High-Variable Genes

We conducted an ablation to determine how gene panel composition influences predictive performance and expert specialization. We trained a baseline model using only Hallmark pathway genes (*Hallmark-Only*) and compared it to our full model trained on the combined *Hallmark+HVG* panel.

To analyze expert specialization, we compared the Jensen-Shannon Distance (JSD) of routing distributions (Figure 7), revealing that the router dynamically aligns its decision with the dominant target gene panel. In the *Hallmark-Only* model (Figure 7a), routing follows *functional semantics*. Since hallmark genes capture broad biological activities, the router groups cancers by shared metabolic states. However, the massive inflammatory signature unique to Lung cancer dominates the proliferation signals (Ridker et al., 2017; Herbst et al., 2018), isolating LUNG as a functional outlier. Conversely, the HVG panel (details in Appendix D.1.1) introduces structural markers often absent from pathway sets, such as Keratins (*KRT5*, *KRT17*) and adhesions (*CEACAM6*). These signals disentangle biologically distinct subclasses, separating primary Breast cancer (IDC) from its lymph node metastasis (LYMPH_IDC) based on tissue-specific expression, with JSD increased from 0.03 to 0.06 (Figure 7b).

### D.1.1. KEY DRIVER GENES IN HVG PANEL

While the full gene panel lists will be available in the code repository, Table 14 highlights the specific high-variance genes that drive the structural shifts in expert routing.

*Table 14.* **Selected High-Variance Genes driving Expert Specialization.** These genes, present in the HVG panel, provide the orthogonal signals necessary for the structural resolution observed in Figure 7.

| MECHANISM | FUNCTION | KEY GENES (FROM HVG PANEL) |
|---|---|---|
| | EPITHELIAL LINEAGE | *KRT5, KRT14, KRT17, EPCAM* |
| **DIFFERENTIATION** | GI TRACT IDENTITY | *CEACAM6, MUC6, TFF3* |
| | IMMUNE MICROENV. | *CD19, MS4A1, CD79A, CD3E* |

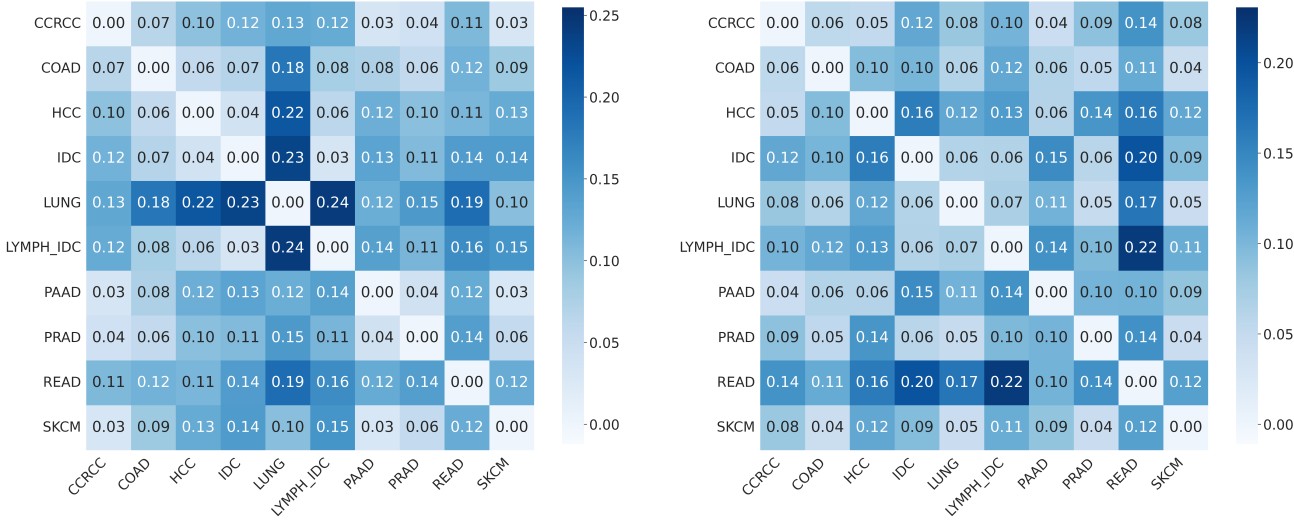

*(a)* **Hallmark-Only**: Functional Alignment

*(b)* **Hallmark+HVG**: Structural Resolution

*Figure 7.* **Impact of Target Gene Panel on Expert Routing.** (a) When restricted to pathway signals, the router preserves the isolation of functionally distinct inputs (LUNG). (b) When exposed to broad genomic signals (HVG), the router adapts to capture granular distinctions (separating IDC from LYMPH_IDC).

*Table 15.* Top50 HVG PCC ↑ across 10 cancer types. Best in **bold**.

| Cancer Type | MLP | STPath | STFlow | STEM | MoLF (Top2of6) | MoLF (Top2of8) |
|---|---|---|---|---|---|---|
| CCRCC | $0.194_{0.012}$ | $0.117_{0.001}$ | $0.128_{0.037}$ | $0.096_{0.028}$ | $\mathbf{0.231}_{0.065}$ | $\underline{0.218}_{0.073}$ |
| COAD | $0.343_{0.114}$ | $0.393_{0.185}$ | $0.310_{0.002}$ | $0.235_{0.076}$ | $\mathbf{0.406}_{0.167}$ | $\underline{0.395}_{0.164}$ |
| HCC | $0.064_{0.045}$ | $0.094_{0.052}$ | $0.081_{0.047}$ | $0.052_{0.015}$ | $\mathbf{0.101}_{0.045}$ | $\underline{0.094}_{0.037}$ |
| IDC | $0.561_{0.125}$ | $\mathbf{0.629}_{0.126}$ | $0.518_{0.078}$ | $0.376_{0.095}$ | $\underline{0.628}_{0.124}$ | $0.611_{0.157}$ |
| LUNG | $0.507_{0.030}$ | $0.518_{0.028}$ | $0.465_{0.040}$ | $0.235_{0.001}$ | $\underline{0.525}_{0.041}$ | $\mathbf{0.531}_{0.002}$ |
| LYMPH_IDC | $0.217_{0.071}$ | $0.182_{0.075}$ | $0.189_{0.045}$ | $0.079_{0.020}$ | $\underline{0.245}_{0.060}$ | $\mathbf{0.248}_{0.060}$ |
| PAAD | $0.407_{0.122}$ | $\mathbf{0.493}_{0.100}$ | $0.416_{0.122}$ | $0.283_{0.151}$ | $\underline{0.460}_{0.084}$ | $0.410_{0.125}$ |
| PRAD | $\underline{0.335}_{0.065}$ | $0.257_{0.012}$ | $0.227_{0.005}$ | $0.197_{0.074}$ | $\mathbf{0.348}_{0.003}$ | $0.322_{0.006}$ |
| READ | $0.264_{0.040}$ | $\mathbf{0.280}_{0.030}$ | $0.226_{0.056}$ | $0.168_{0.013}$ | $\underline{0.275}_{0.045}$ | $0.269_{0.045}$ |
| SKCM | $0.515_{0.104}$ | $0.588_{0.113}$ | $0.557_{0.087}$ | $0.237_{0.139}$ | $\underline{0.606}_{0.127}$ | $\mathbf{0.637}_{0.069}$ |
| **Average** | $0.341_{0.160}$ | $0.361_{0.072}$ | $0.312_{0.168}$ | $0.196_{0.004}$ | $\mathbf{0.382}_{0.173}$ | $\underline{0.373}_{0.178}$ |

## D.2. Qualitative Visualization of Expert Specialization

For each of the six experts in our model, we identified the top 36 patches in Figure 8 from the entire pan-cancer test set that yielded the highest gating logit score.

## E. Appendix: MoE Hyperparameter Ablation

While exhaustive hyperparameter tuning regarding the number of experts may yield marginal gains, our selected configuration (top-2 routing with 6 experts) serves as a robust proof-of-concept that consistently outperforms existing baselines across our comprehensive evaluation. To further validate this architectural choice, we provide additional ablation studies on the number of experts across various benchmarks.

As shown in Tables 15 and 16, the model demonstrates high stability in predicting both highly variable genes and hallmark pathways. Furthermore, Table 17 highlights the model's zero-shot generalization capabilities, confirming that MoLF maintains superior performance and remains resilient to moderate variations in the MoE configuration.

*Table 16.* Model performance comparison measured in PCC of variance-stratified hallmark genes. Best in **bold**.

| MODEL | LOW-VAR ↑ | MID-VAR ↑ | HIGH-VAR ↑ | OVERALL AVG. ↑ |
|---|---|---|---|---|
| MLP | $0.225_{0.028}$ | $0.144_{0.018}$ | $0.145_{0.143}$ | $0.171_{0.045}$ |
| STFLOW | $0.158_{0.011}$ | $0.100_{0.007}$ | $0.107_{0.003}$ | $0.122_{0.029}$ |
| STPATH | $0.177_{0.016}$ | $0.113_{0.011}$ | $0.137_{0.013}$ | $0.143_{0.031}$ |
| MOLF (TOP2OF6) | $\mathbf{0.242}_{0.003}$ | $\mathbf{0.159}_{0.001}$ | $\mathbf{0.167}_{0.006}$ | $\mathbf{0.185}_{0.002}$ |
| MOLF (TOP2OF8) | $\underline{0.227}_{0.000}$ | $\underline{0.147}_{0.002}$ | $\underline{0.157}_{0.000}$ | $\underline{0.174}_{0.000}$ |

*Table 17.* Zero-shot cross-species inference on mouse melanoma. Performance measured in PCC ↑. Best in **bold**.

| MODEL | PCC ↑ |
|---|---|
| MLP | $0.145_{0.009}$ |
| STPATH | $0.090_{0.000}$ |
| STFLOW | $0.126_{0.023}$ |
| MOLF - SKIN CANCER ONLY | $0.127_{0.047}$ |
| MOLF (TOP2OF6) | $\underline{0.202}_{0.001}$ |
| MOLF (TOP2OF8) | $0.205_{0.008}$ |

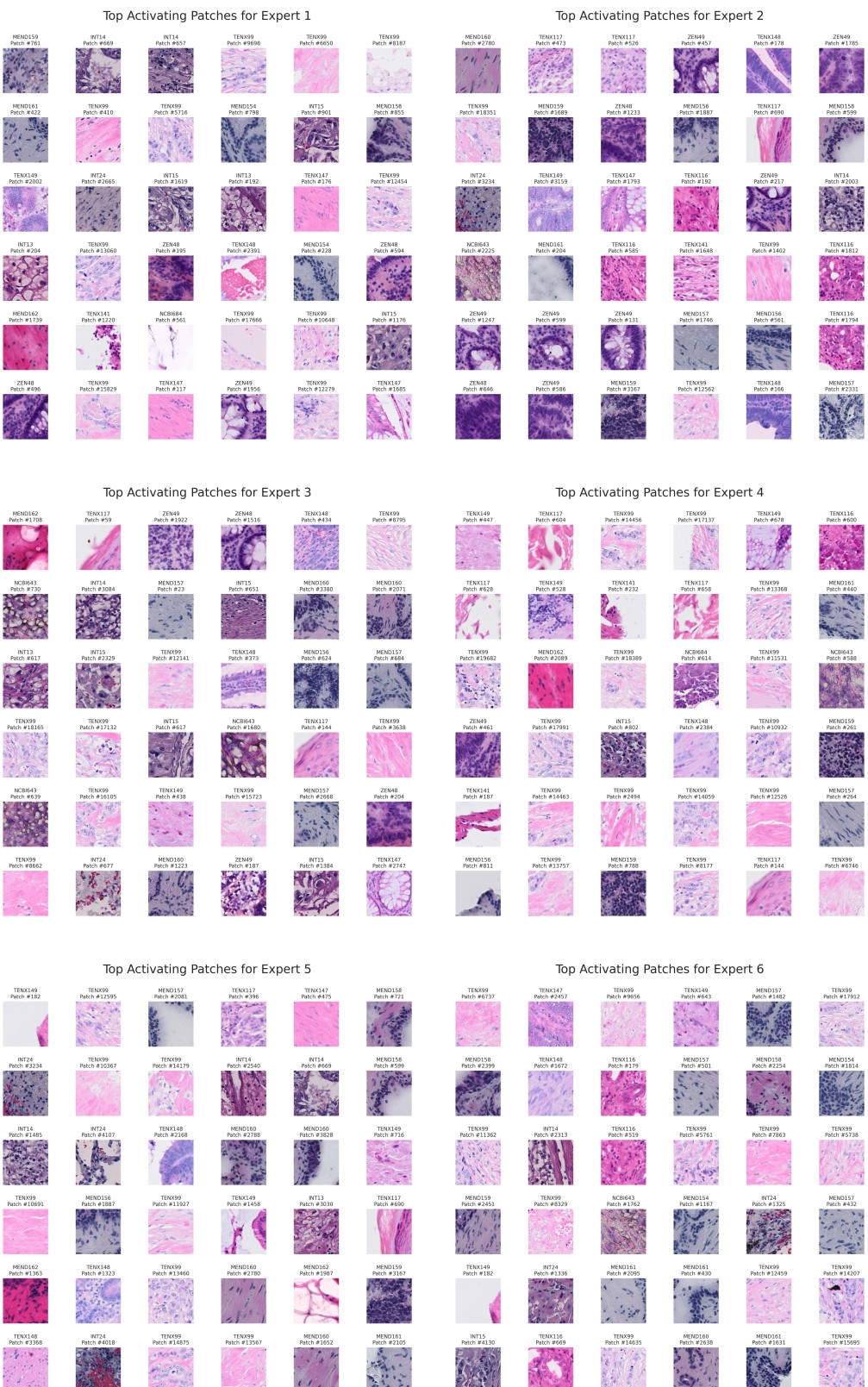

*Figure 8.* Montage of six experts' most activated patches.

