# OpenReview forum: "MoLF: Mixture-of-Latent-Flow for Pan-Cancer Spatial Gene Expression Prediction from Histology"
_ICML.cc/2026/Conference — ICML 2026 regular_

### Official Review · Reviewer_YK3b · 2026-03-03

**Soundness:** 3
**Presentation:** 2
**Significance:** 3
**Originality:** 2
**Overall Recommendation:** 4
**Confidence:** 3

**Summary:**

This paper studies the inference of spatial transcriptomics from histopathology images. To this end, the authors propose a generative model built upon conditional flow match framework, in which they introduce a composite velocity mechanism to capture the diversity and conflicts originated from the pan-cancer heterogeneity. Comprehensive experiments have been conducted to validate the effectiveness of the proposed method. The authors claim that their method achieves new SOTA.

**Compliance With Llm Reviewing Policy:**

Affirmed.

**Final Justification:**

The authors have addressed most of my concerns and I maintain my score as the overall recommendation.

**Key Questions For Authors:**

1. Could the authors provide a detailed description of the histopathology image preprocessing pipeline? For example, how are the whole-slide images processed (e.g., tiling strategy, resolution selection, stain normalization, quality control)?

2. Could the authors provide spatial heatmaps of the predicted gene expression overlaid on the corresponding histopathological slides?

**Limitations:**

Since the method targets biological and medical applications, it would be helpful to include a dedicated discussion on potential biases and risks.

**Strengths And Weaknesses:**

**Strengths:**
1. This paper addresses a timely and highly compelling problem in computational pathology, as there is growing interest in uncovering associations between morphological characteristics and molecular profiles.


2. The authors conduct comprehensive experiments for validation with a synthetic dataset and a real-world benchmark. I particularly appreciate the experiment of expert specialization and utilization that justifies the motivation of the proposed method.



**Weaknesses:**
1. The methodology is not described clearly enough. For example, how is the cross-attention layer that combines cancer-type and morphological features implemented? What are the architectural details of the gating network in Eq. (7)? How is the load-balancing loss computed? The missing mathematical details hider a clear understanding of the proposed methodology.


2. Although I appreciate the authors provide the section Geometric Analysis of Generative Transport for qualitative assessment of the flow matching procedure, the presentation remains confusing. For example, the authors claim that the left panel of Figure 3 shows Gaussian noise projected in UMAP space. If it is the case, why are there points labeled as prediction and ground truth in this panels? What the gray points denote?  Moreover, adding systematic and quantitative evaluations would make the analysis more convincing and better support the claims regarding the effectiveness of the proposed method.


3. As the core originality of the paper lies in the dynamic velocity composition, a thorough justification of its design is desired. In particular, how is the method affected by the number of expert networks and the selection parameter $k$? A systematic analysis of these factors would help clarify the robustness and practical applicability of the proposed approach.


4. There are a few typos in equations. For example, the symbol theta should be a subscript in Eq. (8). The notation $\psi$ is used inconsistently between Eq. (2) and Eq. (5).

---

> ### Author Rebuttal · Authors · 2026-03-27
>
> Thank you for your careful review, the positive assessment of our experimental validation, and for recognizing the significance of this problem. We address your specific questions and methodological details below:
>
> - **Weakness 1**: the cancer type is one-hot encoded and concatenated with the morphology features, serving as the combined input to the cross-attention layer. Furthermore, *as briefly noted at Line 154, we adopted the load-balancing loss from [1]* and *will add the following detailed mathematical formulation to the revised manuscript to eliminate any ambiguity*: For a batch of $M$ samples, let the routing probability distribution over $N$ experts be $p\_{m,i} = \\frac{\\exp(h\_{m,i})}{\\sum\_{j=1}^N \\exp(h\_{m,j})}$, where $h\_m$ represents the gating logits for $m$-th sample. We define $f\_i = \\frac{1}{M} \\sum\_{m=1}^{M} p\_{m,i}$ as the mean routing probability allocated to expert $i$, and $P\_i = \\frac{1}{M} \\sum\_{m=1}^{M} \\mathbb{1}\\{i \\in \\mathrm{TopK}(h\_m, k)\\}$ as the fraction of samples routed to expert $i$ using routing assignments. The auxiliary load-balancing loss, which is minimized when samples are distributed uniformly across all experts, is defined as the scaled dot product of these distributions: $\\mathcal{L}\_{\\mathrm{aux}} = N \\sum\_{i=1}^{N} f\_i \\cdot P\_i$. Finally, this is integrated into our total training objective as $\\mathcal{L}\_{\\mathrm{total}} = \\lambda\_{\\mathrm{flow}}\\mathcal{L}\_{\\mathrm{CFM}} + \\lambda\_{\\mathrm{gene}} \\mathcal{L}\_{\\mathrm{gene}} + \\lambda\_{\\mathrm{aux}} \\mathcal{L}\_{\\mathrm{aux}}$.
>
> - **Weakness 2**: We apologize for the confusing labeling in Figure 3. To clarify: the gray points represent the 2D UMAP projection of the latent gene manifolds for all test cancer types. The red points at $t=0$ represent randomly initialized Gaussian noise (we will explicitly rename this in the revision). After one integration step, the predicted velocity field transports this noise to overlap with the target latent positions (blue). Regarding quantitative evaluation: **comprehensive, systematic quantitative results are provided in the main text tables**. *The geometric transport visualization serves as an intuitive, qualitative supplement to those robust metrics.*
>
> - **Weakness 3**: While more extensive hyperparameter tuning over the number of experts may yield marginal improvements, our chosen configuration (top-2 routing with 6 experts) already provides **a strong proof-of-concept that consistently outperforms existing baselines under comprehensive evaluation**. We further **include an additional ablation** on *the number of experts*, which shows that our model maintains superior performance despite moderate variation across configurations. *We leave exhaustive architectural grid search to future work*. **Please refer to the tables in the rebuttal for reviewer AUg2 and the baselines are omitted due to character limit.** Sorry for the inconvenience.
>
> - **Weakness 4**: thanks for the notation check. We will correct these in the revision.
>
> - **Question 1**: Regarding preprocessing: *histopathology image preprocessing was natively performed and standardized by the authors of the HEST-1k dataset*. Rather than processing raw whole-slide images, we directly utilized the HEST-1k authors' pre-computed, spot-centric 224x224 pixel patches. The dataset providers extracted these patches centered precisely on the spatial transcriptomic spots at 20x magnification, a standardized resolution that perfectly matches the native pre-training parameters of the UNI2 [2] foundation model. For morphological feature extraction (Line 200), we utilized the pathology foundation model UNI2 [2]. To ensure optimal feature extraction, we strictly adhered to their standard pretraining parameters, resolution selections, and normalization.
>
> - **Question 2**: We could additionally visualize prominent marker heatmaps and compare them to published biological findings; however, such biological validation is somewhat beyond the immediate scope of model selection. If the reviewers find it valuable, we are happy to add an appendix section with selected heatmaps and brief literature-based validation.
>
> - **Limitations**:We agree that discussing the clinical implications of this work is important. We will expand the Limitations section to explicitly discuss the potential biases and risks inherent in applying generative models to medical applications.
>
> [1] Fedus, William, Barret Zoph, and Noam Shazeer. "Switch transformers: Scaling to trillion parameter models with simple and efficient sparsity." Journal of Machine Learning Research 23.120 (2022): 1-39.
>
> [2] Chen, Richard J., et al. "Towards a general-purpose foundation model for computational pathology." Nature medicine 30.3 (2024): 850-862.

---

> > ### Author Rebuttal · Reviewer_YK3b · 2026-04-02
> >
> > Thank you for your response to my comments.
> >
> > Regarding your reply to **Weakness 2**, *I could not identify any quantitative analysis corresponding to Figures 3 and 4*. For instance, could the alignment be evaluated quantitatively? Moreover, rather than relying on selected points as illustrative examples in the figures, quantitative results would provide a more comprehensive view of the overall performance.
> >
> > Regarding your reply to **Question 2**, I believe the authors should indeed provide such heatmaps. Certain marker genes are of particular biological importance, and visualizing their spatial patterns would help demonstrate whether the model captures meaningful molecular signals. In addition, overall evaluation metrics alone may not adequately reflect the model’s predictive performance on individual key genes. Even when the overall performance is satisfactory, it is still necessary to examine whether the method can reliably predict biologically important marker genes. Strong performance on these markers would further suggest that the proposed method may be useful not only for gene expression prediction, but also for tasks such as genetic alteration analysis and cancer molecular subtype identification. Therefore, this type of biological validation would be valuable and should not be considered entirely beyond the scope of the present work.

---

> > > ### Author Response · Authors · 2026-04-02
> > >
> > > Thank you for your continued engagement. We address your follow-up points below:
> > >
> > > - **Weakness 2**: *We highly appreciate your insightful suggestion to quantify the flow trajectory itself, as analyzing these intermediate generative dynamics is a fascinating direction*. However, because there is **no intermediate "ground truth" during this continuous flow**, the trajectory cannot currently be directly quantified. Instead, **the quantitative validation of this alignment relies on the accuracy of the final generated state.** This is precisely what our *Pearson Correlation Coefficient (PCC) metrics* on *HVGs* and *stratified Hallmark gene sets* demonstrate in the main tables. The figures serve as an intuitive, qualitative supplement to these rigorous PCC metrics. We will explicitly clarify this link in the revised captions.
> > >
> > > - **Question 2**: While the primary focus of our work remains on computational methodology and model evaluation, we understand your perspective that visualizing specific marker genes offers valuable supplementary biological context. The text-based rebuttal interface prevents us from attaching images here, but *we are happy to accommodate this request in the revised manuscript*. We will include a dedicated Appendix section featuring spatial heatmaps of selected marker genes, side-by-side comparisons with ground-truth spatial transcriptomics.

---

### Official Review · Reviewer_AUg2 · 2026-03-09

**Soundness:** 3
**Presentation:** 3
**Significance:** 2
**Originality:** 2
**Overall Recommendation:** 4
**Confidence:** 5

**Summary:**

The paper proposes a novel method for gene expression prediction from histology images using a latent flow matching approach conditioned on the image representation and cancer type. Specifically, the novelty of the methods lies in leveraging a mixture of experts architecture for the latent flow matching models. The model is evaluated on the HEST1k benchmark datasets, consisting of a pan-cancer collection of spatial transcriptomics datasets, on 2 different splits and using a novel gene panel selection.

**Compliance With Llm Reviewing Policy:**

Affirmed.

**Final Justification:**

The authors have addressed my questions, and I appreciate the reply. I would like to use this space to quickly comment on the last point they made about the "global context" v. MoE representation learning tradeoffs. I disagree with describing passing the cancer label to the adaaptive conditioning of the generative model as global context, it's just a label info that steers the generative model and makes it easier to sample in that region of the space. I do agree though that the MoE could indeed pick up more fine grained structure that nevertheless occurs pan-cancer. The authors provided great examples of the type of signal the MoE could indeed pick up that type of variation. But they should show it! It would be a really great showcase of their model interpretability power.

Nevertheless, the paper have merits and I have increased my score.

**Key Questions For Authors:**

- The macro-micro transport figures to highlight interpretability of the model results are interesting, albeit a bit confusing:  in the macro case, what does it mean that at t=0 the points are perfectly clustered an overlayed over the UMAP, if they represent the gaussian noise stage of the ODE? Are those samples from the guassian prior and not the latent flow matching prior? please explain.
- The paper highlights the usage of MoE architecture, enabling the model to resolve "conflicting signal" during pan cancer training. However, the velocity vector field network is conditioned entirely on the cancer type, so how are the experts able to route cancer-specific signals if the conditioning is applied at the input, for all experts simultaneously? An interesting ablation would be to remove the cancer-type conditioning and evaluate performance in an OOD setting, such as for the results reported in Table 5.

**Limitations:**

yes

**Strengths And Weaknesses:**

The paper proposes a novel architecture for a latent flow matching model applied to the generation of gene expression from imaging, in the context of spatial transcriptomics. The paper is clear and well-written, and the experiments are clearly motivated.
Strength:
- The pan cancer approach of training the flow matching across cancer conitioning jointly on the image representation and the cancer type.
- The OOD evaluation in Table 5 is a strong result, and highlights the benefit of the pan cancer training.

Weaknesses:
- The experiments of table 1 are only representative of 2 splits, making it hard to properly evaluate model performance, the authors should aim to present the experiment results for at least one more fold split for fairer comparison.
- More ablation on the architecture itself (number of experts, routing strategy etc.) would be beneficial, and ideally they would run on a real world dataset, and not synthetic dataset. Also, why is the decoder-level loss required during the flow matching training? An ablation result would strengthen the claim.
- Originality: the paper presents a 2-stage latent flow matching approach applied to spatial transcriptomics data. The application is reasonable but relatively straightforward, since all the building blocks are well established.

---

> ### Author Rebuttal · Authors · 2026-03-27
>
> Thank you for acknowledging the strength of our out-of-distribution evaluations and the benefits of our pan-cancer approach. We address your specific concerns and questions below:
>
> - **Weakness 1**: We restricted our evaluation to the established **HEST-Bench holdout test set** (a subset from HEST1 curated by the authors themselves) to ensure a rather *fair comparison* with prior state-of-the-art baselines (e.g., STPath). Because we strictly enforce mutually exclusive test samples across splits to avoid uninsightful more splits, and *some cancer types in HEST-Bench contain very few samples*, **2 is the mathematical maximum number of non-overlapping splits possible**. These two splits adequately capture the available variance within the strict constraints of this standardized benchmark.
>
> - **Weakness 2**: To clarify, **MoLF is evaluated on real-world data**. **HEST-1k** is a large-scale, real-world clinical dataset, not a synthetic one. *Regarding the architecture*: While more extensive hyperparameter tuning over the number of experts may yield marginal improvements, our chosen configuration (top-2 routing with 6 experts) already provides **a strong proof-of-concept that consistently outperforms existing baselines under comprehensive evaluation**. We further **include an additional ablation** on *the number of experts*, which shows that our model maintains superior performance despite moderate variation across configurations. *Baselines are omitted due to character limit*. Sorry for the inconvenience. *We leave exhaustive architectural grid search to future work*. Regarding the **decoder loss during flow matching**: it acts strictly as an auxiliary supervised signal to regularize the latent flow matching trajectory. *The decoder weights remain fully frozen during this stage*. This formulation ensures the generated latents remain aligned with the target decoding manifold, *a practice analogous to Eq. 2 in STFlow* where they also include this supervised MSE loss between generated gene expression and target gene expression at flow matching stage.
>
> **Table 1: Top-50 HVG PCC ↑ across 10 cancer types (MoLF only)**
>
> | Cancer Type | MoLF (Top2of6) | MoLF (Top2of8) |
> |------------|----------------|----------------|
> | CCRCC | **0.231±0.065** | _0.218±0.073_ |
> | COAD | **0.406±0.167** | _0.395±0.164_ |
> | HCC | **0.101±0.045** | _0.094±0.037_ |
> | IDC | _0.628±0.124_ | 0.611±0.157 |
> | LUNG | _0.525±0.041_ | **0.531±0.002** |
> | LYMPH_IDC | _0.245±0.060_ | **0.248±0.060** |
> | PAAD | _0.460±0.084_ | 0.410±0.125 |
> | PRAD | **0.348±0.003** | 0.322±0.006 |
> | READ | _0.275±0.045_ | 0.269±0.045 |
> | SKCM | _0.606±0.127_ | **0.637±0.069** |
> | **Average** | **0.382±0.173** | _0.373±0.178_ |
>
> **Table 2: PCC of variance-stratified hallmark genes (MoLF only)**
>
> | Model | Low-Var ↑ | Mid-Var ↑ | High-Var ↑ | Overall Avg. ↑ |
> |------|-----------|-----------|------------|----------------|
> | MoLF (Top2of6) | **0.242±0.003** | **0.159±0.001** | **0.167±0.006** | **0.185±0.002** |
> | MoLF (Top2of8) | _0.227±0.000_ | _0.147±0.002_ | _0.157±0.000_ | _0.174±0.000_ |
>
> **Table 3: Zero-shot cross-species inference (mouse melanoma, MoLF only)**
>
> | Model | PCC ↑ |
> |------|-------|
> | MoLF (Top2of6) | _0.202±0.001_ |
> | MoLF (Top2of8) | **0.205±0.008** |
>
> - **Weakness 3**: While flow matching and Mixture of Experts (MoE) exist independently, their **integration to handle the highly complex, multi-modal distribution of pan-cancer spatial transcriptomics is novel**. For example, one of our baseline STFlow is accepted at ICML 2025. Their building blocks flow matching, graph neural network, and frame averaging are also well established. But their novelty in accurately assembling those building blocks for solving complex biology problems is acknowledged.
>
> - **Question 1**: In the macro-transport visualization, the Gaussian noise initialization is **not** perfectly aligned with the latent UMAP, but *the UMAP is the maximum space that the warped gaussian noise can scatter on*. There are empty parts on the UMAP. It will be clearer if we reduce the size of red dots. We chose a rather large size for visualization. We shall improve the graphics in the revision.
>
> - **Question 2**: **The experts routing is not conditioned on cancer types**. **It’s learnt dynamically with load-balancing loss**. *We do not intend to hard-route experts with cancer types*. The cancer type conditioning happens at the cross attention layer which projects the image feature into lower dimension for flow matching condition. *Regarding the cancer type conditioning*, when a pathologist reviews an H&E case, the primary tumor type (e.g., breast carcinoma) is routinely provided in the metadata. Leveraging this aligns with standard practice in the field (e.g., STPath). Furthermore, to handle edge cases where this data is missing, **we explicitly include an "unknown" cancer type in training**, which we *successfully leverage for zero-shot cross-species inference* in section 4.5.

---

> > ### Author Rebuttal · Reviewer_AUg2 · 2026-04-01
> >
> > Thank you for the exhaustive rebuttal and comments to all my points. I also appreciate the ablation study on testing different mixtures. Reagrding the conditional vector field with mixture of experts architecture, I don't understand your answer and would like to clarify my question again: if the vector field is conditioned on the cancer/tissue type, how is it possible for the the MoE to reconcile "conflicting signal" from the pan-cancer training? I understand it probably serves as some sort of object-level regularization that aims at capturing the same axis of variation across cancer types, but I would like to know more what is your intuition about what kind of signal the MoE actually captures, given that the vector field itself is conditioned on the cancer type. I am overall satisfied with the rebuttal and will increase my score.

---

> > > ### Author Response · Authors · 2026-04-02
> > >
> > > Thank you for your very quick follow-up, your engagement with our ablations, and your willingness to increase your score.
> > >
> > > Your comment elegantly described our architecture. However, we find it helpful to decouple the two mechanisms you mentioned, "*object-level regularization*" and "*capturing shared axes of variation*", into the two levels:
> > > 1. **Conditioning provides global context and object-level regularization**: Because Pathology Foundation Models (PFMs) are trained in a self-supervised manner without explicit labels, they lack inherent disease context. Injecting the cancer type via cross-attention provides this missing global context. This makes the model more "morphology-aware," highlighting the visual features most relevant to that specific disease without erasing shared features.
> > > 2. **MoE resolves local morphological conflicts**: Even after this regularization, local tissue architecture (e.g., blood vessels, stroma, or immune infiltrates) varies across a single slide. Furthermore, similar-looking local structures in different organs might express genes differently. The MoE resolves these downstream conflicts by capturing the "shared axes of variation" as you framed. It specializes in mapping these specific local tissue patterns to their target gene expressions across different tissues.
> > >     - This is why experts do not simply specialize by cancer type (Section 4.4), allowing the model to generalize to rare or unknown diseases better.
> > >     - Furthermore, Appendix D.2 confirms that specific gene panels drive expert routing behavior, proving the experts are learning local visual-to-gene mappings rather than just acting as a secondary tissue classifier.
> > >
> > > Interpretability: We agree that fully mapping what these MoE experts learn into human-readable concepts is a challenge, which we note as a limitation. Future work will bridge this gap, potentially by using Vision-Language Models (VLMs) to automatically describe the specific image patches that activate each expert.

---

### Official Review · Reviewer_MYeC · 2026-03-12

**Soundness:** 2
**Presentation:** 2
**Significance:** 2
**Originality:** 2
**Overall Recommendation:** 3
**Confidence:** 4

**Summary:**

This paper proposes MoLF, a two-stage generative framework for predicting spatial transcriptomics from H&E histology across multiple cancer types. Stage I learns a latent gene manifold via a Transformer VAE, Stage II uses conditional flow matching with a MOE architecture conditioned on image features and cancer type to predict gene expression. The paper claims SOTA on the HEST-1k pan-cancer benchmark and zero-shot cross-species transfer to mouse melanoma.

**Compliance With Llm Reviewing Policy:**

Affirmed.

**Final Justification:**

Thanks to the authors for the rebuttal. My main concern about adding additional benchmarks against regression based models has been addressed, and I have updated my scores accordingly.

**Key Questions For Authors:**

For MoE ablation, is the dense model parameter size aligned with MoE model?

**Limitations:**

the authors adequately discussed the limitations

**Strengths And Weaknesses:**

- Strength

  - The cross-species transfer result is interesting.

- Weakness
  - The overall methodological contribution feels somewhat limited with a MoE + conditional flow matching model. While this is a reasonable engineering combination, the paper does not fully establish that this constitutes a substantial advance beyond prior flow-based or mixture-based modeling ideas.
  - A central conceptual concern is that the model is explicitly conditioned on cancer type via a one-hot label in addition to image features. This makes the pan-cancer problem easier. In many cases, coarse cancer identity may already be inferable from pathology alone, especially with richer whole-slide context, so the real challenge may be better visual modeling rather than explicit subtype injection.
  - The evaluation misses the comparison with stronger deterministic regression models, MLP is weak.

minor: I think authors mean  log1p on line 57.

---

> ### Author Rebuttal · Authors · 2026-03-27
>
> Rebuttal
> Thank you for your review and for highlighting the strength of our zero-shot cross-species transfer results. We address your specific concerns below:
> - **Weakness 1**: While flow matching and Mixture of Experts (MoE) exist independently, their **integration to handle the highly complex, multi-modal distribution of pan-cancer spatial transcriptomics is novel**. For example, one of our baseline STFlow is accepted at ICML 2025. Their building blocks flow matching, graph neural network, and frame averaging are also well established. But their novelty in accurately assembling those building blocks for solving complex biology problems is acknowledged. MoLF systematically **outperforms the standard flow-based baseline (STFlow)** across all evaluations. To the best of our knowledge, **MoLF is the first mixture-based modeling approach** for pan-cancer spatial transcriptomics prediction by the time of our submission.
>
> - **Weakness 2**:  Regarding the cancer type conditioning: we condition on cancer type because it mirrors real-world clinical workflows. When a pathologist reviews an H&E case, the primary tumor type (e.g., breast carcinoma) is routinely provided in the metadata. **Leveraging this aligns with standard practice in the field (e.g., STPath)**. Furthermore, to handle edge cases where this data is missing, **we explicitly include an "unknown" cancer type in training**, which we *successfully leverage for zero-shot cross-species inference in section 4.5*.
>
> - **Weakness 3**: *Recent state-of-the-art generative frameworks (e.g., STPath, STFlow, STEM) have already comprehensively demonstrated that generative modeling systematically outperforms deterministic regression approaches* (such as TRIPLEX [1], BLEEP [2], STNet [3], and HisToGene [4]) in this domain. Because the superiority of generative models over these regression baselines is already well-established in the recent literature, we focused our evaluation on comparing MoLF against the strongest available generative SOTA. *The MLP was included solely as a minimal reference baseline*.
>
> - **Key Questions For Authors**:  The MoE model utilizes 6 one-layer transformer experts with a model dimension of 256 and 4 attention heads as stated in appendix B. The dense baseline model was scaled up to a one-layer transformer of dimension 768 with 8 attention heads. So dense model actually has slightly higher parameter count. But now we also added an ablation with 8 experts, so **the MoE model and dense model are aligned in parameter size**.
>
> Thank you for catching the typo; we will correct this to log1p on line 57 in the revision.
>
> [1]Chung, Youngmin, et al. "Accurate spatial gene expression prediction by integrating multi-resolution features." Proceedings of the IEEE/CVF Conference on Computer Vision and Pattern Recognition. 2024.
>
> [2] Xie, Ronald, et al. "Spatially resolved gene expression prediction from histology images via bi-modal contrastive learning." Advances in Neural Information Processing Systems 36 (2023): 70626-70637.
>
> [3] He, Bryan, et al. "Integrating spatial gene expression and breast tumour morphology via deep learning." Nature biomedical engineering 4.8 (2020): 827-834.
>
> [4]Pang, Minxing, Kenong Su, and Mingyao Li. "Leveraging information in spatial transcriptomics to predict super-resolution gene expression from histology images in tumors." BioRxiv (2021): 2021-11.

---

> > ### Author Rebuttal · Reviewer_MYeC · 2026-04-03
> >
> > Thanks authors for the detailed response. While my concerns are addressed, for Weakness 3, I do believe the paper can benefit from comparisons with stronger regression-based methods.

---

> > > ### Author Response · Authors · 2026-04-04
> > >
> > > **Thank you for confirming that your concerns are fully resolved.**
> > >
> > > While we agree that broader baseline comparisons are generally beneficial, *we also believe in building upon established consensus to push the field forward*. Because recent literature (e.g., STPath [npj digital medicine 2025], STFlow [ICML 2025], STEM [ICLR 2025]) has already conclusively proven the superiority of generative models over deterministic regression methods, we leveraged those established findings.
> > >
> > > This allowed us to focus our resources entirely on the competitive frontier, the available generative SOTA, rather than reproducing historical regression baselines.
> > >
> > > Thank you again for your time and constructive engagement with our work.

---

### Official Review · Reviewer_V2CT · 2026-03-15

**Soundness:** 2
**Presentation:** 3
**Significance:** 3
**Originality:** 2
**Overall Recommendation:** 3
**Confidence:** 4

**Summary:**

This paper studies the problem of predicting spatial gene expression from histology images in a pan-cancer setting, where a single model is trained across multiple cancer types. The authors propose MoLF, a generative modeling framework that first learns a latent gene manifold using a Transformer-based VAE and then models the conditional distribution of gene expression using a conditional flow matching process. The velocity field of the flow model is parameterized with a Mixture-of-Experts (MoE) architecture intended to better capture heterogeneous tissue patterns across cancers.

**Compliance With Llm Reviewing Policy:**

Affirmed.

**Final Justification:**

The rebuttal addresses part of my original concerns, particularly by clarifying the evaluation setup and providing additional justification for the gene-panel choice and the MoE ablation. However, I still think a direct sensitivity analysis with different HVG thresholds would be important to verify robustness to the specific gene selection criterion used in the main evaluation, so my overall evaluation remains unchanged.

**Key Questions For Authors:**

1) Why are only 10 cancer types evaluated when the HEST-1k dataset contains 25 cancer types?
2) How sensitive are the results to the Top-50 HVG gene selection?
3) Since the model already conditions on cancer type through the context vector, how much of the reported improvement can be attributed specifically to the MoE architecture rather than conditional modeling? A clearer ablation isolating these factors would help clarify the mechanism behind the reported gains.
4) Reproducibility of the proposed framework may be challenging without access to the implementation details. Do the authors plan to release the code to reproduce experimental results?

**Limitations:**

No. The paper briefly acknowledges the limited interpretability of expert specialization, but it does not sufficiently discuss several other important limitations. In particular, it does not address the evaluation on only a subset of cancer types from the HEST-1k dataset, the reliance on a Top-50 HVG gene panel, the difficulty of attributing the reported gains to the MoE architecture rather than explicit cancer-type conditioning, or the computational cost and scalability of the proposed framework. A more complete discussion of these issues would strengthen the paper.

**Strengths And Weaknesses:**

**Soundness**

The proposed architecture is technically reasonable and builds on established techniques such as VAEs, conditional flow matching, and Mixture-of-Experts models. The idea of using MoE to address heterogeneity across cancer types is plausible. However, several aspects of the experimental design weaken the strength of the empirical evidence supporting the central pan-cancer modeling claim.

First, the experiments in Table 2 evaluate only 10 cancer types, while the HEST-1k dataset contains 25 cancer types. The paper does not explain how this subset was selected or why the remaining cancer types were excluded. Since the motivation of the work is to model cross-cancer heterogeneity, restricting the evaluation to a subset of cancer types without justification makes it difficult to assess whether the method generalizes across the full dataset.

Second, the reported improvements are primarily reflected in average performance across cancer types, while several individual cancer types still achieve better results with other baselines (e.g., STPath). A more detailed analysis of per-cancer performance differences would help clarify whether the method consistently improves modeling across heterogeneous cancer types.

Third, the evaluation focuses on the Top-50 Highly Variable Genes (HVGs). While HVGs provide strong signal, restricting evaluation to such a small subset may introduce potential selection bias. Sensitivity analysis with different HVG thresholds (e.g., Top-100 or Top-200) would help determine whether the improvements are robust to gene selection choices.

Fourth, the evaluation does not analyze predictive performance across genes outside the HVG subset. Since many genes have lower variance and different statistical properties, analyzing prediction accuracy across the broader gene distribution would provide a more complete assessment of model utility.

Finally, it is unclear whether the Mixture-of-Experts architecture is necessary to address the heterogeneity problem. The model already conditions explicitly on cancer type through the context vector, which may already capture much of the cross-cancer variation. Additional comparisons with simpler conditional architectures (e.g., a single conditional flow model without MoE) would help clarify whether the performance gains are specifically due to the MoE design. The expert specialization analysis in Figure 5 also provides limited evidence that MoE effectively decomposes heterogeneous patterns. Expert activations appear broadly distributed across the latent space rather than forming clearly specialized regions, making it unclear whether the experts capture distinct sub-distributions of the data.

**Presentation**

The paper is generally well organized and the overall modeling pipeline is described clearly. The figures help illustrate the architecture and generative process. However, several aspects of the presentation could be improved. Key experimental design decisions—such as the selection of cancer types and the effect of the gene panel—are not clearly explained in the main text.

**Significance**

Predicting gene expression from histology images is an important problem in computational pathology and spatial transcriptomics. A reliable histology-to-expression model could potentially enable large-scale molecular profiling without requiring expensive spatial transcriptomics experiments. The idea of training unified models across multiple cancer types is also meaningful, as pan-cancer models may leverage shared biological structure across tissues.

However, the current empirical evidence does not clearly demonstrate that the proposed architecture substantially advances the ability to model cross-cancer heterogeneity beyond existing conditional modeling approaches. Broader evaluation across cancer types and gene sets would strengthen the practical significance of the work.

**Originality**

The paper builds on a line of recent work that already explores flow-matching-based generative modeling for histology-to-expression prediction. In this context, the main methodological addition here appears to be the introduction of a mixture-of-experts velocity field for pan-cancer modeling.

While this is a reasonable extension, the originality of this addition is not yet clearly established. In particular, the paper does not cleanly isolate whether the reported gains come from the MoE design itself, from explicit conditioning on cancer type, or from evaluation choices such as the selected cancer-type subset and the Top-50 HVG gene panel. Since these factors are all entangled in the current experiments, it is difficult to determine what new modeling capability is specifically introduced by the MoE component.

As a result, the paper’s originality appears to lie in extending an existing flow-matching framework with MoE for this application setting, but the evidence is currently insufficient to clearly demonstrate that this extension provides a distinct and well-justified methodological advance beyond prior work.

---

> ### Author Rebuttal · Authors · 2026-03-27
>
> Thank you for the review! Here we address your key questions:
> - Q1: We evaluated our model on the 10 cancer types from **the standardized HEST-Bench** (*a curated evaluation subset of the HEST-1k dataset by authors themselves*). *For training we include all available slides covering 29 cancer types including "unknown"*. We chose this specific benchmark to ensure a rigorous and fair comparison with prior state-of-the-art models, such as STPath, STEM, and STFlow, which also base their evaluations on HEST-Bench or its subsets. Specifically, *using the established HEST-Bench holdout test set was necessary to conduct a direct comparison with STPath, for which we only have inference access*.
> - Q2: As high-variance signals dominate gradient propagation, HVGs are comparatively easier to learn. We therefore additionally include Hallmark gene sets to evaluate performance on more stable and less variance-driven biological signals. (**detailed in Lines 210-224**). Hallmark gene sets cover a wider variance level.These two settings probe complementary capabilities: capturing strong, cancer-specific variability (HVGs) and modeling conserved biological programs (Hallmarks). Consistent performance across both demonstrates that our results are not overly sensitive to the specific gene selection criterion. Corresponding results are reported in **Tables 3, 7, and 9**, and the gene variance distribution is shown in **Appendix Figure 6**.
> - Q3: We have already isolated these factors in our ablations. **Table 6** (Section 4.6.1) *explicitly compares the model with and without the MoE architecture, while keeping the cancer-type conditioning constant*. **The performance gain shown in this table is strictly attributable to the MoE design, independent of the conditioning**. Regarding the *cancer type conditioning*: we condition on cancer type because it *mirrors real-world clinical workflows*. When a pathologist reviews an H&E case, the primary tumor type (e.g., breast carcinoma) is routinely provided in the metadata. Leveraging this aligns with standard practice in the field (e.g., STPath). Furthermore, **to handle edge cases where this data is missing, we explicitly include an "unknown" cancer type in training**, which we successfully leverage for *zero-shot cross-species inference* in section 4.5.
> - Q4: Yes, we will release the code upon acceptance **as stated in line 580**. We are fully committed to open science.
>
> - Limitations: In our revision, we will incorporate the clarifications provided above regarding our evaluation setup (HEST-Bench standard). Gene panel curation (robustness across HVG and Hallmark sets in Lines 210–224), and the MoE ablation results (Table 6) are already in the paper. Additionally, we will explicitly add a discussion on computational cost and scalability. We will note that the training is done on a single A100 GPU.

---

> > ### Author Rebuttal · Reviewer_V2CT · 2026-04-04
> >
> > The rebuttal addresses part of my original concerns, particularly by clarifying the evaluation setup and providing additional justification for the gene-panel choice and the MoE ablation. I appreciate the authors' effort in responding carefully.
> >
> > However, I still think a direct sensitivity analysis with different HVG thresholds (e.g., Top-100 or Top-200) would be important to verify that the reported improvements are robust to the specific gene selection criterion used in the main evaluation.

---

> > > ### Author Response · Authors · 2026-04-04
> > >
> > > Thank you for your continued engagement and for acknowledging our previous clarifications.
> > >
> > > While we completely agree that exhaustive ablations are generally valuable for model profiling, **we respectfully posit that testing Top-100 or Top-200 HVGs would not yield new methodological insights beyond our existing evaluation**, for both conceptual and practical reasons:
> > >
> > > 1. **Orthogonal Validation** vs. **Redundant Bias**: **Expanding the threshold from Top-50 to Top-100 or Top-200 simply adds genes further down the exact same variance-ranked tail**. To rigorously verify our model is not overly sensitive to high-variance genes, we deliberately used the **Hallmark gene sets**. Because Hallmarks are curated based on conserved biological functions rather than mathematical variance, **they provide a much stronger, completely orthogonal test of robustness**.
> > >
> > > 2. **Comprehensive Statistical Coverage**: As demonstrated in our gene variance distribution (**Appendix Figure 6**), *the Hallmark sets naturally capture a much broader and more representative statistical distribution than simply sliding an HVG threshold*.
> > >
> > > 3. **Computational Scope and Established Proof-of-Concept**: From a practical standpoint, running additional ablation sweeps across our entire rigorous evaluation pipeline (which already requires training our model plus 3 baselines across two splits) is prohibitive within the rebuttal timeframe. Because **our dual-panel approach (HVGs + Hallmarks)** already firmly *establishes and validates the core methodological concept of this paper*, we respectfully leave exhaustive incremental threshold sweeps for future work.
> > >
> > > We hope this clarifies why our current experimental design provides a comprehensive and scientifically rigorous proof of robustness.

---

### Decision · Program_Chairs · 2026-04-30

**Decision:**

Accept (regular)

**Comment:**

Reviewers found the problem timely and of importance. Predicting gene expression from histology images is an important problem in computational pathology and spatial transcriptomics. The method is relatively straightforward, since all the building blocks are well established. Reviewers appreciated both the pan cancer approach of training the flow matching across cancer conditioning jointly on the image representation and the cancer type and the experimental evaluation.

The rebuttal addressed reviewers concerns. Remain concerns from the two WR reviewers were answered by the authors in their finalk comments.

Overall this paper advances the state of the art in this problem and should be published if there is space in the program